# Changes in anxiety and depression levels and meat intake following recognition of low genetic risk for high body mass index, triglycerides, and lipoproteins: A randomized controlled trial

Ga Young Lee[1], Kyong-Mee Chung[2], Junghak Lee[3], Jeong-Han Kim[3], Sung Nim Han[1,4]*

1 Department of Food and Nutrition, College of Human Ecology, Seoul National University, Seoul, Korea, 2 Department of Psychology, Yonsei University, Seoul, Korea, 3 Department of Agricultural Biotechnology and Research Institute of Agriculture and Life Sciences, Seoul National University, Seoul, Korea, 4 Research Institute of Human Ecology, Seoul National University, Seoul, Korea

* snhan@snu.ac.kr

**Data Availability Statement:** All relevant data are within the paper and its Supporting Information files.

## Abstract

### Background

Psychological status affects dietary intake, and recognizing genetic information can lead to behavior changes by influencing psychological factors such as anxiety or depression.

### Objectives

In this study, we examined the effects of disclosing genetic information on anxiety or depression levels and the association between these psychological factors and dietary intake.

### Methods

A total of 100 healthy adults were randomly assigned to an intervention group (n = 65) informed about their genetic test results regarding body mass index and lipid profiles (triglyceride and cholesterol concentrations) and a not-informed control group (CON, n = 35). Based on polygenic risk scores, participants in the intervention group were subclassified into an intervention-low risk (ILR, n = 32) and an intervention-high risk (IHR, n = 33) group. Nutrient and food intakes were assessed via a 3-day dietary record at baseline and at 3 and 6 months. Depression and anxiety levels were measured using PHQ-9 and GAD-7 questionnaires, and the relative levels of blood metabolites were measure using GC-MS/MS analysis.

### Results

Noticeable changes in dietary intake as well as psychological factors were observed in male subjects, with those perceiving their genetic risks as low (ILR) showing a significant increase in protein intake at 3 months compared to baseline (ILR: 3.9 ± 1.4, p<0.05). Meat intake also

**Funding:** This research was awarded to SNH and supported by the grant from the Seoul National University Research Grant in 2018 (350-20180049). The funder, Seoul National University, had no role in study design, data collection and analysis, decision to publish, or preparation of the manuscript.

**Competing interests:** The authors have declared that no competing interests exist.

**Abbreviations:** BMI, body mass index; TG, triglyceride; LDL cholesterol, low-density lipoprotein cholesterol; HDL cholesterol, high-density lipoprotein cholesterol.

increased significantly in males in the ILR group at 3 months, but not in the IHR and CON groups (ILR: 49.4 ± 30.8, IHR: -52.2 ± 25.4, CON: -5.3 ± 30.3 g/d). ILR group showed a significant decrease in anxiety levels at 3 months, and their anxiety scores showed a negative association with meat intake (standardized $\beta$ = -0.321, $p<0.05$). The meat intake at 3 months was associated with the relative levels of arginine and ornithine (standardized $\beta$ = 0.452, $p<0.05$ and standardized $\beta$ = 0.474, $p<0.05$, respectively).

## Conclusions

Taken together, anxiety levels were decreased in male subjects who perceived their genetic risk to be low, and the decrease in anxiety levels was associated with an increase in meat intake. This suggests that recognizing genetic information may affect psychological factors and dietary intake.

## Introduction

Psychological factors can play motivational or regulatory roles in dietary choices [1–3]. It has been demonstrated that psychological conditions such as high levels of stress, depression, and anxiety are likely to increase appetite and are strongly associated with poor diet quality due to consumption of foods and beverages with high energy density and sugar content [4–8].

The use of genetic testing in the general population and clinical practice has increased, because it provides personalized information and recommendations regarding health and lifestyle behaviors at a relatively low cost [9]. Previous studies have shown that disclosing genetic information has the potential to motivate individuals to manage their health by inducing positive changes in health-related behaviors, which can lead to weight loss, qualitative improvements in dietary fat consumption, and reductions in sodium intake [10–15]. However, it has also been reported that recognizing genetic information can have adverse psychological effects on individuals and can cause or exacerbate anxiety or depression, depending on the degree of perception and the severity of the perceived risks [16–19].

A number of studies have investigated the effect of genetic information recognition on changes in anxiety and depression levels, but existing studies demonstrated that there was limited evidence to support the effect of genetic risk disclosure on psychological changes [20,21]. However, different psychological responses have been reported according to the severity of diseases and heterogeneous study designs including measurement tools for psychological changes [18,20]. In addition, there are few studies that investigated the effect of providing genetic information regarding general wellness such as body mass index (BMI) or lipid profiles. Overall, the impact of genetic information disclosure on psychological factors is not well characterized and is still needed to be elucidated.

While there is a number of studies that investigated whether disclosing genetic information is associated with changes in dietary intake [10,13,14,22,23], conflicting results have been reported. A recent systematic review of 11 randomized controlled trials reported a greater improvement in dietary intake in subjects who received personalized nutritional advice based on genetic information compared to those who only received generalized dietary advice [24]. On the other hand, in a systematic review that evaluated dietary intake changes as outcome variables including nutrients, food, and dietary supplements, only six out of 18 studies (33%) reported positive and health-promoting effects regarding dietary changes [25].

Although changes in dietary intake or anxiety and depression levels following the disclosure of genetic information have been examined in several studies, few have investigated the effects on both anxiety and depression levels and dietary choices. Therefore, here, we aimed to investigate whether recognizing genetic information regarding personal risks for a high BMI and abnormal triglyceride (TG) and lipoprotein levels affects anxiety or depression levels, and whether changes in anxiety and depression are accompanied by changes in nutrient or food intakes.

## Materials and methods

### Study design and subjects

This study is a single-blinded randomized, parallel-group trial with a 2 (intervention):1 (control) subject ratio. Subjects were adults free of diseases, aged 25 to 35 years, and with a BMI of 18.5 to 25 kg/m$^2$. They were enrolled from December 2018 to January 2019 through poster advertisements. A total of 111 applicants who expressed their intention to participate in the study were screened for qualification, and 100 eligible subjects provided written consents. The study was conducted between February and September 2019. It received approval from the Seoul National University Institutional Review Board (IRB #1901/001-004) and was registered in the Clinical Research Information Service database (CRIS; KCT0004650).

### Procedure

Subjects were randomly assigned to the intervention (n = 65) and the control (CON, n = 35) group (2:1 ratio) because the intervention group consisted of those who could have "risk" or "no-risk" genotypes of genes in genetic testing results. Randomization was conducted using a random sequence generated with PROC SURVEYSELECT in SAS version 9.4 (SAS Institute, Cary, NC, USA). The random sequence was concealed until the baseline survey was completed. The intervention group was further subclassified into an intervention-low risk (ILR, n = 32) and an intervention-high risk (IHR, n = 33) group, based on the polygenic risk score calculated for each gene item by assigning 1 point for "borderline risk" and 2 points for "caution". The subjects in the ILR and the IHR group were informed about their genetic test results regarding the risks for a high BMI and abnormal lipid profiles, such as elevated TG and cholesterol concentrations, at the beginning of the intervention, whereas those in the CON group were not informed about their test results until the end of the study. No further treatment was provided. While the researcher, which is the outcome assessor, was blinded to the group assignment information by using randomly assigned IDs for all subjects used in both online and document-type questionnaires, all the study subjects were exposed to the random assignment results.

### Direct-to-consumer (DTC) genetic testing

Genotyping was conducted by the TheragenEtex Bio Institute (TheragenEtex Inc., Suwon, Korea), on peripheral blood collected from subjects at the Seoul National University Health Service Center at the baseline visit. The DTC test report included information on the integrated risk represented as "Good" for not having genotype at risk, "Borderline risk" for carrying one genotype at risk, and "Caution" for having two genotypes at risk regarding each gene item, that is, BMI, TG concentrations, and low-density (LDL) and high-density lipoprotein (HDL) cholesterol levels, as well as recommendations for those at risk (S1 Table).

## Study outcomes

The primary outcome of the study was a change in dietary intake including nutrients and servings of each food group from baseline to each follow-up assessment following the subject's recognition of their genetic risk information regarding their BMI, TG, and lipoprotein levels. The secondary outcome was the change in anxiety and depression levels, that is, anxiety and depression symptom levels measured with the Generalized Anxiety Disorder 7-item scale (GAD-7) and the Patient Health Questionnaire 9-item scale (PHQ-9), from baseline to each follow-up assessment.

**Anthropometric assessment.** Anthropometric measurements were conducted at baseline and at the 3- and 6-month follow-ups. Height (cm) and weight (kg) were measured using a digital stadiometer (BSM 330, Biospace Co. Ltd., Seoul, Korea), and body fat (kg), waist-to-hip ratio (%), and skeletal muscle mass (kg) were determined using a bioelectrical impedance analyzer (Inbody 720, Biospace Co. Ltd.).

**Assessment of nutrient and food intake.** Nutrient and food intakes were determined with a dietary record for a total of 3 days (2 weekdays and 1 weekend day) at baseline and at the 3- and 6-month follow-ups. The dietary records submitted by the subjects were reviewed by a clinical dietitian on the day of the visit to ensure accuracy. The nutrient analysis was performed using the Computer Aided Nutritional Analysis Program (CAN-Pro) 5.0 (web version; The Korean Nutrition Society, Korea).

Consumed foods were classified into 23 food groups according to the 2020 Dietary Reference Intakes for Koreans [26]. The classification included seven grain categories, six protein categories, three vegetable categories, two fruit categories, one dairy category, one fat category, two added sugar categories, and one fast food category (S2 Table).

**Assessment of anxiety and depression levels.** The subjects completed online self-report surveys at the baseline visit and at the 3- and 6-month follow-ups using SurveyMonkey (http://www.surveymonkey.com). The surveys were created based on the GAD-7 and the PHQ-9 (S3 Table). The GAD-7 is a seven-item, self-administered rating scale that evaluates anxiety symptoms, with an overall score ranging from 0 to 21. It has been reported that the GAD-7 has a good internal consistency expressed as Cronbach's α in Korean version of validation study was 0.92 [27], and the Cronbach's α in this study was 0.91. The PHQ-9 is a nine-item, self-administered rating scale that evaluates depressive symptoms; its total score ranges from 0 to 27. The PHQ-9 is widely used to test symptoms of depression in medical or primary care settings [28–30]. The internal consistency of PHQ-9 reported in the Korean version validation study was 0.95 [31], and Cronbach's α was 0.88 in this study.

**Physical activity (PA) measurement.** The evaluation of PA was conducted using online survey site SurveyMonkey (http://www.surveymonkey.com) based on the International Physical Activity Questionnaire-short form (IPAQ-SF) at the baseline and the 3-, and 6-month follow-ups (S4 Table). Subjects were instructed to record the duration, frequency, and intensity of their physical activities. Data obtained from the IPAQ-SF was transformed into continuous variables following the IPAQ-SF analysis guideline, and expressed as Metabolic Equivalent of Task (MET) value-hours per week. The total MET score represents the sum of all activity MET values (8.0 for Vigorous activity, 4.0 for Moderate activity, 3.3 for Walking activity).

**Measurement of metabolites by GC-MS/MS.** Serum samples for metabolite analysis were obtained from peripheral blood collected at the Seoul National University Health Care Center at the baseline, and at 3 and 6 months. The metabolites were analyzed using GC-MS/MS (Shimadzu GCMS-TQ8040) with multiple reaction monitoring ions at the Pesticide Chemistry and Toxicology laboratory of the Department of Agricultural Biotechnology of Seoul National University. The analytical conditions for serum metabolite analysis using

GC-MS/MS were previously described in a study [32]. In summary, the GC-MS/MS parameters were as follows: injector temperature = 250˚C, ion source and transfer line temperatures = 200˚C, 280˚C. The initial oven temperature was set to 60˚C for 2 min, then increased to 320˚C at a rate of 10˚C/min, held for 15 min. Helium carrier gas flow rate = 1 mL/min, electron ionization energy = 70 eVA. Out of the 304 metabolites, 93 metabolites including isomers or various derivatives were statistically analyzed. Following peak confirmation, the peak area of the metabolites was compared to that of the internal standard 2-isopropylmatic acid to calculate the relative area under the curve for each metabolite.

## Statistical analysis

We estimated the sample size by using a 1-year trial design similar to this study design [14]. Using the difference in dietary fat quality of 1.9 points between the high-apolipoprotein E genetic-risk group and low-risk group, the standard deviation of 3.6, and a power of 80% resulted in a calculated sample size of 55 per group. Considering the loss to follow-up, 65 participants were recruited for the intervention group. All analyzes were performed using the intent-to-treat principle. The differences in changes in nutrient and food intakes and anxiety or depression levels among the groups were analyzed using ANOVA for normally distributed data and Kruskal-Wallis tests for skewed data, followed by Bonferroni tests for multiple comparisons for post-hoc analysis. The changes in nutrient intakes and anxiety or depression levels were assessed using paired $t$ tests for normally distributed data and Wilcoxon signed-rank tests for skewed data. Chi-square tests were used to compare frequencies of categorical variables among the groups. A Spearman correlation was used to assess the relationship between nutrients and meat intake. Multivariate linear regression analyses were performed to determine the association between the anxiety or depression levels and dietary intakes. Age, physical activities, and energy intake were used as covariates. Analyses were conducted using SPSS Statistics version 26 (IBM SPSS Statistics, Chicago, IL, USA) and statistical significance was set at $p < 0.05$ for two-sided tests.

## Results

### Characteristics of the study subjects

Of all 100 subjects, 89 (56 in the intervention group and 33 in the CON group) completed the study (Fig 1). The baseline characteristics of all study subjects are shown in Table 1. The mean age of the study subjects was 28.1 ± 2.1 years, and the mean BMI was 22.3 ± 2.0 kg/m². The average GAD-7 score was 3.7 ± 3.5, and the average PHQ-9 score 5.1 ± 4.6. There were no significant differences among the CON, ILR, and IHR groups regarding sex, age, anthropometric measures, PA, and nutrient intake. However, the PHQ-9 score was significantly higher in the ILR than in the CON group at baseline.

### Changes in dietary intake

Changes in energy (kcal/d) and fat (% energy/d) intakes from baseline after the disclosure of genetic information to the participants in the intervention groups did not differ among groups. Carbohydrate intake (% energy/d) had increased significantly at 3 months in the CON group compared to baseline ($p < 0.05$), but no significant differences were found in the ILR and IHR groups. On the other hand, a significant increase in protein intake (% energy/d) was observed in the ILR group at 3 months compared to baseline ($p < 0.05$), while no significant changes were observed in the CON and IHR groups. The magnitude of change in protein intake was significantly higher in the ILR group at 3 months than in the IHR group (Table 2). Male

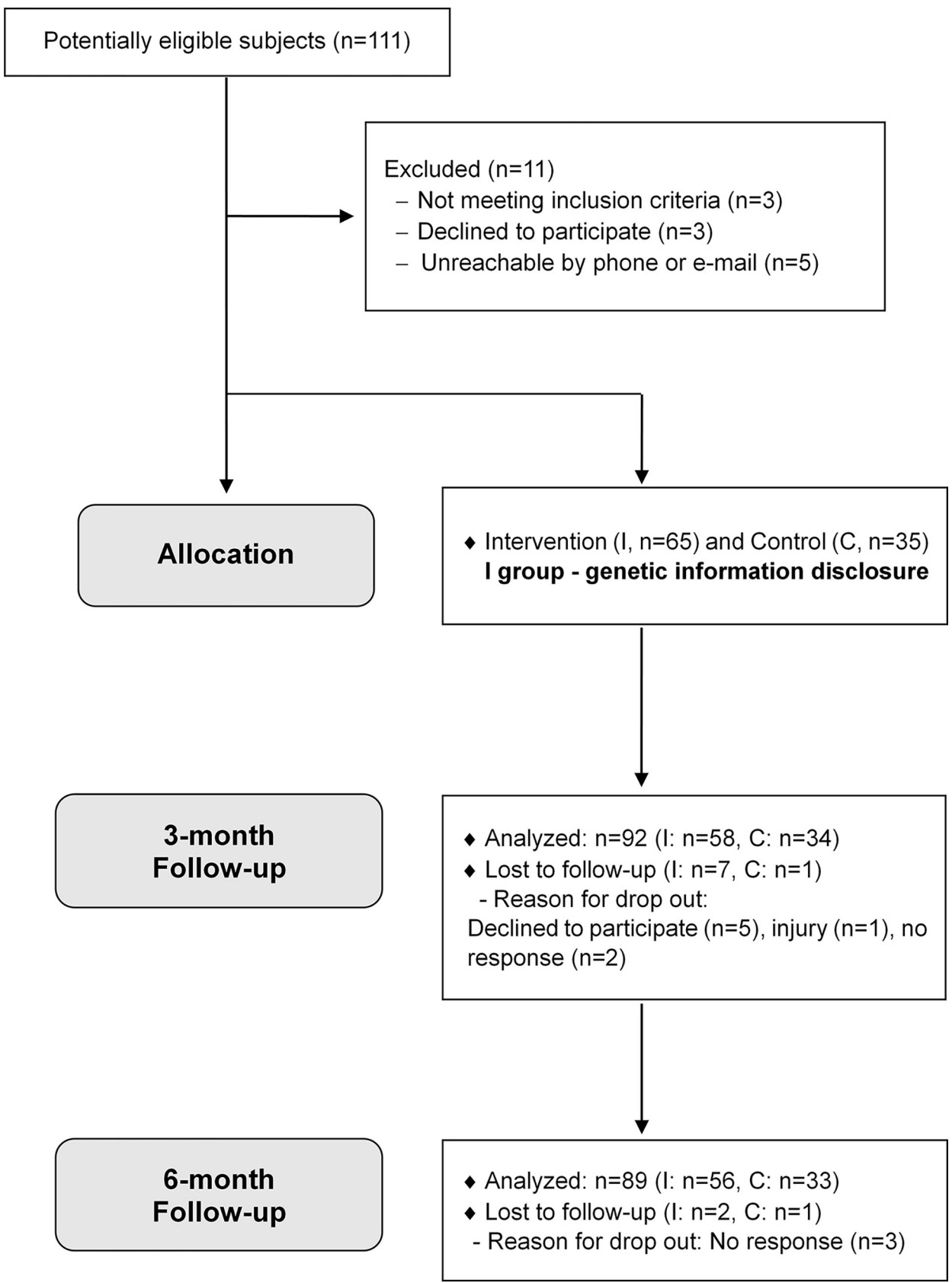

**Fig 1. Flow diagram of the study.**

**Table 1. Baseline characteristics of the study subjects[a].**

| | Total (*n* = 100) | CON (*n* = 35) | ILR (*n* = 32) | IHR (*n* = 33) |
|---|---|---|---|---|
| **Sex n (%)** | | | | |
| Men | 50 (50) | 17 (48.6) | 16 (50) | 17 (51.5) |
| Women | 50 (50) | 18 (51.4) | 16 (50) | 16 (48.5) |
| **Age (years)** | 28.1 (2.1) | 27.9 (1.7) | 27.9 (2.3) | 28.5 (2.4) |
| **Education n (%)** | | | | |
| High school | 5 (5) | 3 (8.6) | 2 (6.2) | 0 (0) |
| Undergraduate degree | 34 (34) | 12 (34.3) | 11 (34.4) | 11 (33.3) |
| Graduate school or above | 61 (61) | 20 (57.1) | 19 (59.4) | 22 (66.7) |
| **Mean (SD)** | | | | |
| **Weight (kg)** | 63.7 (10.6) | 64.0 (10.1) | 63.0 (12.0) | 64.0 (9.9) |
| BMI (kg/m$^2$) | 22.3 (2.0) | 22.4 (1.9) | 21.7 (2.2) | 22.7 (1.8) |
| **Body fat mass (kg)** | 15.6 (3.6) | 16.3 (3.4) | 14.8 (3.6) | 15.9 (3.8) |
| **Waist-to-hip ratio (%)** | 0.9 (0.0) | 0.9 (0.0) | 0.8 (0.0) | 0.8 (0.0) |
| **Skeletal muscle mass (kg)** | 26.7 (6.2) | 26.4 (6.1) | 26.7 (6.7) | 26.9 (6.1) |
| Total PA (MET-hrs/wk) [a] | 34.3 (22.7) | 31.9 (20.4) | 37.0 (25.8) | 34.2 (22.1) |
| **Energy (kcal/d)** | 1888.6 (489.1) | 1934.8 (425.9) | 1809.2 (480.8) | 1916.5 (560.4) |
| **Carbohydrates (% energy/d)** | 47.8 (8.0) | 49.2 (6.7) | 48.9 (8.1) | 45.2 (8.8) |
| **Protein (% energy/d)** | 16.3 (3.7) | 15.8 (3.3) | 16.1 (3.5) | 17.0 (4.3) |
| **Fat (% energy/d)** | 32.5 (5.9) | 32.5 (5.5) | 32.7 (6.0) | 32.2 (6.2) |
| **Total cholesterol (mg/dL)** | 186.2 (28.2) | 189.2 (28.5) | 178.3 (18.0) | 190.6 (34.5) |
| **LDL-cholesterol (mg/dL)** | 100.3 (24.2) | 102.5 (22.2) | 95.7 (20.8) | 102.4 (29.0) |
| **HDL-cholesterol (mg/dL)** | 68.6 (15.1) | 70.2 (16.4) | 65.8 (14.7) | 69.7 (14.1) |
| **Triglyceride (mg/dL)** | 86.7 (43.6) | 82.9 (46.6) | 84.7 (34.3) | 92.8 (48.7) |
| **PHQ-9 score** | 5.1 (4.6) | 4.5 (5.0) | 6.8 (5.2) | 4.2 (2.9) |
| **GAD-7 score** | 3.7 (3.5) | 3.2 (3.2) | 4.3 (4.4) | 3.5 (2.6) |

[a] MET: Metabolic equivalent task.

CON, control; ILR, intervention-low risk; IHR, intervention-high risk; BMI, body mass index; PA, physical activity; PHQ-9, Patient Health Questionnaire 9-item scale; GAD-7, Generalized Anxiety Disorder 7-item scale.

subjects in the ILR group showed no significant change in carbohydrate intake, but their protein intake had increased at 3 months compared to baseline ($p < 0.05$) (Fig 2A and 2C). Female subjects in the CON group showed a significant increase in carbohydrate intake at 3 months and a significant increase in protein intake at 6 months compared to baseline, while no changes were observed in the ILR and IHR groups (Fig 2B and 2D).

There were no significant group differences in the intake of grains, vegetables, dairy products, fats and oils, and beverages or fast food in either male or female subjects. In male subjects, the ILR and IHR groups tended to differ in their magnitude of change in meat intake (g/d) at 3 months ($p = 0.056$) (Fig 3A), with the ILR group showing an increase in meat intake, while that of the IHR group decreased. No significant change was observed in the CON group. The magnitude of change in fruit intake (g/d) significantly differed among groups at 3 months in male subjects only (Fig 3C), with the IHR group showing a significant decrease in fruit intake at 3 months compared to baseline ($p < 0.05$). Male subjects of the IHR group showed a significant increase in the intake of added sugar (g/d), in the form of sugar, fruit juice, and sugar-sweetened beverages, at 6 months compared to baseline (Fig 3E). In female subjects, the consumption of meat, fruit, and added sugar did not differ among groups (Fig 3B, 3D and 3F, S5 Table).

**Table 2. Differences in nutrient intake among the CON, ILR, and IHR groups [a, b, c].**

| | CON (n = 35) | | ILR (n = 32) | | IHR (n = 33) | | p value [2)] |
|---|---|---|---|---|---|---|---|
| | Mean (SEM) | Change [1)] | Mean (SEM) | Change [1)] | Mean (SEM) | Change [1)] | |
| **Energy (kcal/d)** | | | | | | | |
| Baseline | 1934.8 (72.0) | | 1809.2 (85.0) | | 1916.5 (97.6) | | |
| 3-month follow-up | 1692.9 (65.3) * | -241.9 (70.0) | 1686.7 (90.6) | -122.4 (105.4) | 1716.4 (81.0) | -200.2 (92.7) | 0.635 |
| 6-month follow-up | 1805.9 (66.0) | -128.9 (71.1) | 1672.1 (91.1) | -137.1 (121.4) | 1769.4 (84.7) | -147.2 (103.0) | 0.991 |
| **Carbohydrates (% energy/d)** | | | | | | | |
| Baseline | 49.2 (1.1) | | 48.9 (1.4) | | 45.2 (1.5) | | |
| 3-month follow-up | 54.9 (1.8) * | 5.6 (2.0) [a] | 47.6 (2.0) | -1.3 (2.3) [a] | 49.4 (2.4) | 4.2 (2.7) [aa] | 0.019 |
| 6-month follow-up | 48.8 (1.4) | -0.5 (1.4) | 50.3 (1.6) | 1.3 (1.9) | 47.6 (1.7) | 2.4 (1.9) | 0.490 |
| **Protein (% energy/d)** | | | | | | | |
| Baseline | 15.8 (0.5) | | 16.1 (0.6) | | 17.0 (0.8) | | |
| 3-month follow-up | 16.4 (0.7) | 0.7 (0.7) [ab] | 18.9 (0.9) * | 2.8 (0.9) [a] | 17.2 (0.8) | 0.1 (1.1) [a] | 0.046 |
| 6-month follow-up | 16.8 (0.5) | 1.1 (0.6) | 16.2 (0.7) | 0.0 (0.9) | 17.2 (0.5) | 0.2 (0.8) | 0.520 |
| **Fat (% energy/d)** | | | | | | | |
| Baseline | 32.5 (0.9) | | 32.7 (1.1) | | 32.2 (1.1) | | |
| 3-month follow-up | 31.8 (1.6) | -0.8 (1.5) | 35.4 (1.3) | 2.7 (1.7) | 34.7 (1.9) | 2.5 (1.9) | 0.052 |
| 6-month follow-up | 33.2 (1.1) | 0.7 (1.0) | 32.2 (1.4) | -0.6 (1.5) | 33.7 (1.4) | 1.5 (1.7) | 0.606 |

[a] Change = nutrient intake at follow-up–nutrient intake at baseline.

[b] Means with different superscripts indicate the significant differences in changes in nutrient intake among the groups, assessed with one-way ANOVA and Kruskal-Wallis tests followed by Bonferroni tests for multiple comparisons.

[c] The asterisk indicates a significant difference

(*: $p < 0.05$) compared to nutrient intake at baseline.

CON, control; ILR, intervention-low risk; IHR, intervention-high risk.

## Changes in anxiety and depression levels

Changes in GAD-7 scores at the 3-month and 6-month follow-ups from baseline did not differ among groups. However, depression levels, measured with the PHQ-9, had significantly decreased in the ILR group at 6 months compared to baseline, although no significant changes were found in the CON and IHR groups (S6 Table).

When anxiety and depression levels were analyzed separately by sex, male subjects in the ILR group showed a significant decrease in GAD-7 scores at 3 months compared to baseline ($p < 0.05$) and a decrease in PHQ-9 scores at the 3- and 6-month follow-ups (both $p < 0.05$) (Fig 4A and 4C); the magnitude of change in PHQ-9 scores at 3 months was significantly greater in the ILR group than in the CON and IHR groups. In female subjects, changes in GAD-7 and PHQ-9 scores did not significantly differ among groups (Fig 4B and 4D).

## Association between nutrient and food intakes

A significantly positive correlation was found between meat intake (g/d) at the 3-month follow-up and the consumption of protein (% energy/d, $r = 0.437$, $p < 0.05$) and nutrients, that is, thiamin ($r = 0.310$, $p < 0.05$) and vitamin B6 ($r = 0.397$, $p < 0.05$) (S7 Table).

## Correlations between anxiety and depression scores and meat intake

Overall, GAD-7 and PHQ-9 scores did not significantly correlate with meat intake (S8 Table). Furthermore, we observed no changes in anxiety or depression levels and meat intake after the

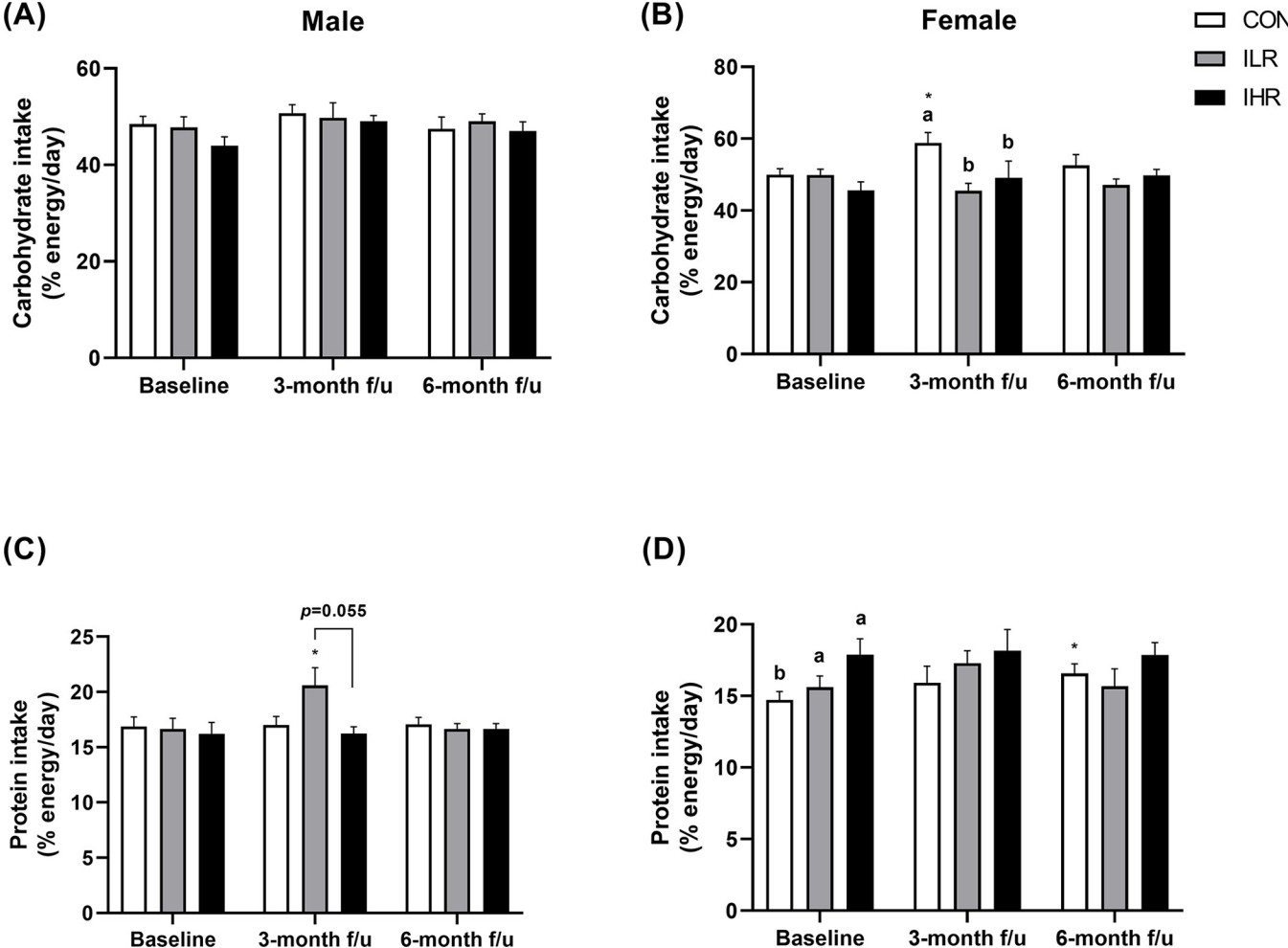

**Fig 2. Differences in carbohydrate and protein intake among the CON, ILR, and IHR groups according to the sex of the study subjects.** Differences in carbohydrate intake among the CON, ILR, and IHR groups in male (A) and female (B) subjects. Differences in protein intake among the groups in male (C) and female (D) subjects. The asterisk indicates significant difference (*: $p<0.05$) compared to the anthropometric measures at the baseline. CON, control; ILR, intervention-low risk; IHR, intervention-high risk.

recognition of the subjects' genetic information in the IHR group, and no significant association between the two indicators was observed. However, subjects in the ILR group showed a significant inverse association between GAD-7 scores and meat intake at 3 months (standardized $\beta$ = -0.364, $p<0.05$), while no significant association was observed between anxiety scores and meat intake at 6 months. The association between anxiety scores at 3 months and meat intake was significant even after controlling for sex, age, BMI, and PA level in the multiple linear regression model (standardized $\beta$ = -0.321, $p<0.05$) (Table 3). There was no significant association between PHQ-9 scores and meat intake (g/d) at either the 3-month or the 6-month follow-up in the ILR group (standardized $\beta$ = -0.219, $p$ = 0.239; standardized $\beta$ = -0.156, $p$ = 0.492, respectively; S9 Table).

## Association between meat intake and blood metabolites

In order to examine the metabolite changes after dietary intake, the relative levels of blood metabolites were measured. Although there were no significant differences observed in the changes of

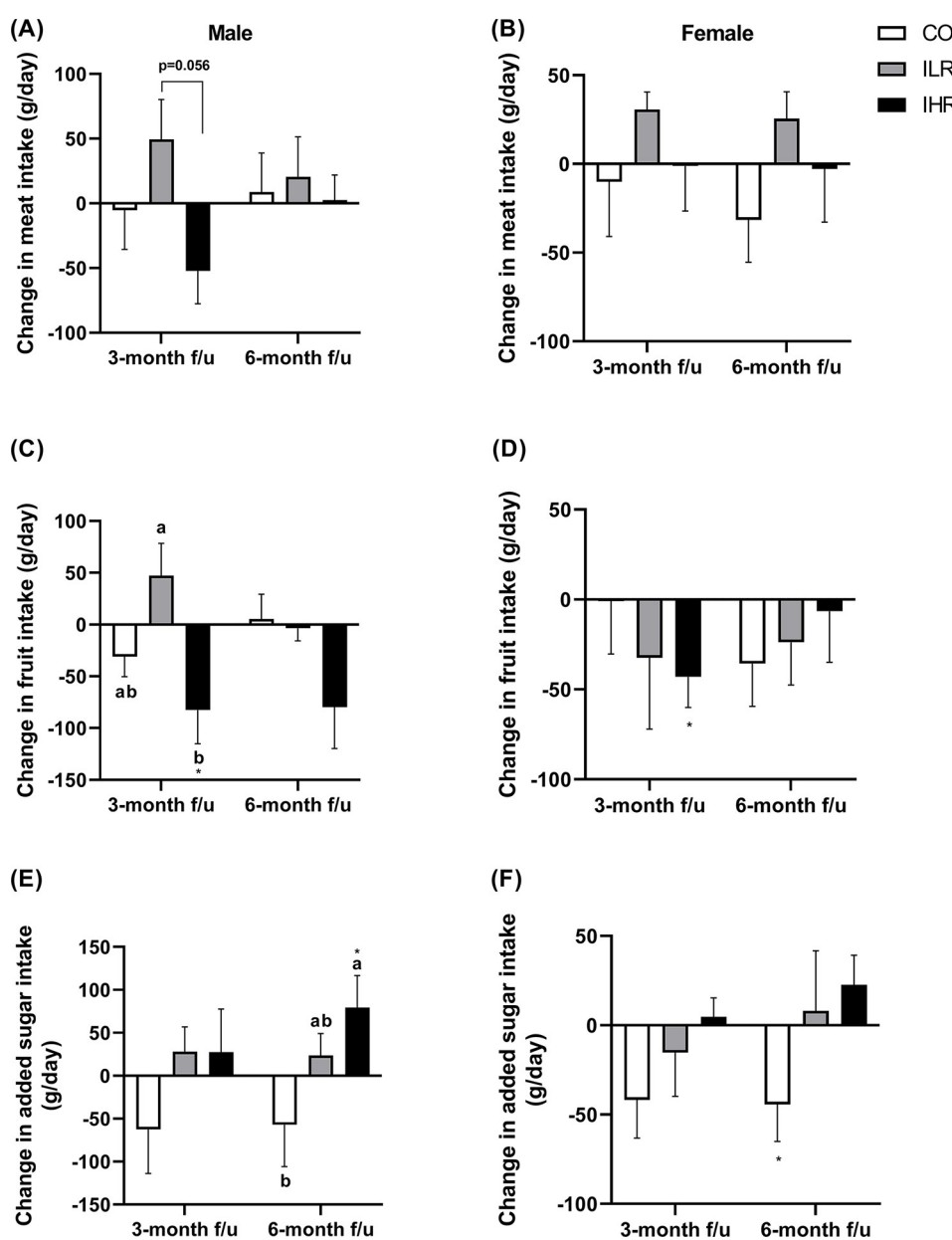

**Fig 3. Changes in the intake of meat, fruit, and added sugar from baseline to 3 and 6 months after subjects were informed about their genetic information.** Changes in the intake of meat (A), fruit (C), and added sugar (E) for all subjects. Changes in the intake of meat (B), fruit (D), and added sugar (F) for male subjects. Means with different superscripts indicate significant differences in changes in food intakes among the groups, tested with one-way ANOVA and Kruskal-Wallis tests followed by Bonferroni corrections for multiple comparisons. The asterisk indicates significant difference (*: $p < 0.05$) in food intake compared to the baseline. CON, control; ILR, intervention-low risk; IHR, intervention-high risk.

arginine and ornithine among groups (S10 Table), a significant positive association was found between meat intake after 3 months and the relative levels of arginine and ornithine among subjects in the ILR group, after controlling for sex, age, BMI, energy intake, and PA level (standardized $\beta = 0.452$, $p < 0.05$ and standardized $\beta = 0.474$, $p < 0.05$) (Table 4). Conversely, no significant association was found between these measures in the CON group or IHR groups.

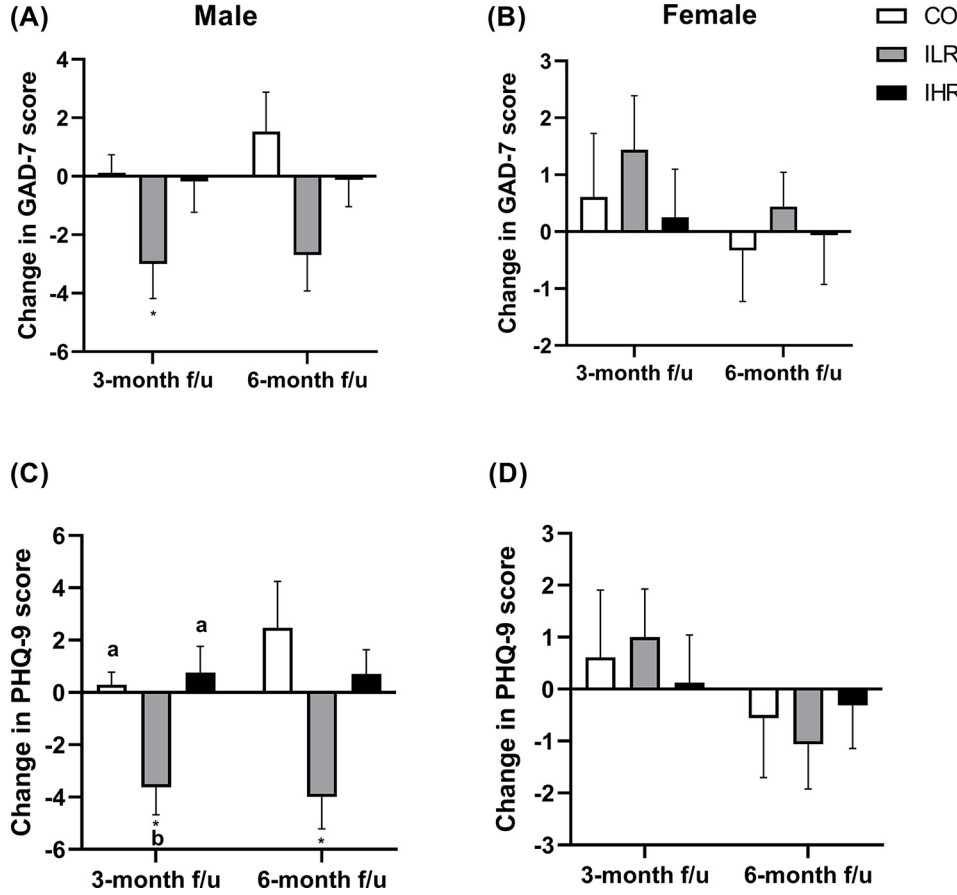

**Fig 4. Changes from baseline to 3 and 6 months in anxiety and depression levels among the CON, ILR, and IHR groups according to the sex.** Means with different superscripts indicate significant differences in changes in GAD-7 (A and B) and PHQ-9 (C and D) scores among the groups, assessed with one-way ANOVA and Kruskal-Wallis tests followed by Bonferroni tests for multiple comparisons. The asterisk indicates significant difference (*: $p<0.05$) in GAD-7 and PHQ-9 scores compared to the baseline. CON, control; ILR, intervention-low risk; IHR, intervention-high risk; GAD-7, Generalized Anxiety Disorder 7-item scale; PHQ-9, Patient Health Questionnaire 9-item scale.

**Table 3. Multiple linear regression analysis on the association between GAD-7 scores and meat intake in the ILR group [a, b].**

| | ILR group (n = 32) | | | | | |
| --- | --- | --- | --- | --- | --- | --- |
| | GAD-7 score at 3-month | | | GAD-7 score at 6-month | | |
| | B (SE) | Standardized $\beta$ | $p$ | B (SE) | Standardized $\beta$ | $p$ |
| **Meat intake (g/d)** | -0.015 (0.007) | -0.321 | 0.048 | -0.010 (0.008) | -0.269 | 0.208 |
| **Women vs. men** | 3.364 (1.621) | 0.390 | 0.048 | 1.065 (1.645) | 0.146 | 0.523 |
| **Age (years)** | 0.795 (0.289) | 0.416 | 0.011 | 0.414 (0.310) | 0.257 | 0.194 |
| **BMI (kg/m²)** | 0.033 (0.377) | 0.016 | 0.931 | -0.225 (0.381) | -0.132 | 0.561 |
| **Total PA (MET-hrs/wk) [a]** | 0.034 (0.026) | 0.201 | 0.204 | -0.006 (0.028) | -0.044 | 0.820 |

[a] MET: Metabolic equivalent task.

[b] The $R^2$ values for GAD-7 scores at 3-month and 6-month were 0.423 and 0.180, respectively.

ILR, Intervention-Low Risk; BMI, body mass index; PA, physical activity; GAD-7, Generalized Anxiety Disorder 7-item scale.

**Table 4. Multiple linear regression analysis on the association between meat intake and blood metabolites in the ILR group [a, b].**

| | ILR group (n = 32) | | | | | |
| --- | --- | --- | --- | --- | --- | --- |
| | Arginine levels at 3-month | | | Ornithine levels at 3-month | | |
| | B (SE) | Standardized $\beta$ | p | B (SE) | Standardized $\beta$ | p |
| Meat intake (g/d) | 0.1 (0.1) | 0.452 | 0.038 | 0.1 (0.04) | 0.474 | 0.023 |
| Women vs. men | 7.4 (9.5) | 0.180 | 0.445 | 5.2 (8.2) | 0.139 | 0.532 |
| Age (years) | 0.9 (1.6) | 0.095 | 0.600 | 1.2 (1.4) | 0.149 | 0.386 |
| BMI (kg/m$^2$) | -0.9 (2.3) | -0.087 | 0.711 | -1.4 (2.0) | -0.159 | 0.474 |
| Energy intake (kcal/d) | 0.004 (0.01) | 0.092 | 0.662 | 0.005 (0.01) | 0.148 | 0.460 |
| Moderate PA (MET-hrs/wk) [a] | 0.05 (0.28) | 0.035 | 0.855 | 0.08 (0.24) | 0.061 | 0.735 |

[a] MET: Metabolic equivalent task.

[b] The R$^2$ values for arginine and ornithine levels at 3-month were 0.260 and 0.338, respectively.

ILR, Intervention-Low Risk; BMI, body mass index; PA, physical activity.

## Discussion

In this study, disclosing genetic information regarding an individual's risk for a high BMI and abnormal TG, LDL, and HDL lipoprotein levels resulted in changes in psychological factors and dietary intake. Depression and anxiety levels were decreased in male subjects who perceived their genetic risk for a high BMI and abnormal lipid profiles to be low, while the intake of meat increased significantly. Furthermore, an inverse association between anxiety levels and meat intake among subjects in the low genetic risk group was observed.

A number of studies have determined changes in dietary intake as a primary outcome regarding the perception of genetic information [24,33–35]. It has been reported that individuals who received personalized advice based on dietary patterns, phenotypes, and genetic information showed significant improvements in Healthy Eating Index-2010 and Mediterranean scores and increased consumption of fruits and vegetables compared to those who only received generalized dietary advice [33,34]. In addition, subjects who were notified of carrying risk alleles for the angiotensin-converting enzyme gene, which is linked to higher sodium sensitivity, showed a decrease in their sodium intake over 12 months compared to a control group [10]. Nevertheless, most of the existing studies provided genetic information related to disease susceptibility to subjects, and few have focused on elucidating the effect of disclosing genetic information regarding general wellness variables, that is, the BMI or lipid profiles, on dietary intake. Large-scale genome-wide association studies have shown that genetic variations in BMI-related genes, such as *FTO*, *BDNF*, and *MC4R*, are closely related to obesity [36–40]. Additionally, single nucleotide polymorphisms that are located at the genetic loci including *GCKR*, *ANGPTL3*, *MLXIPL*, *SORT1*, *HMGCR*, and *ABO* were found to be associated with plasma TG and LDL cholesterol levels [41,42], and high levels of plasma LDL cholesterol and TG are widely known as an independent risk factor for cardiovascular disease [43,44]. BMI and levels of TG and lipoproteins are modifiable factors that can be altered by lifestyle factors such as dietary intake and PA [45]. Therefore, informing people about their genetic information regarding health-related variables such as BMI, TG, and lipoproteins is more likely to motivate them to make changes in health-related behaviors than just providing them with information about specific disease risks.

In this study, protein intake increased significantly in male subjects who perceived their genetic risk to be low at 3 months compared to baseline, while the control group and those who perceived their risk as high showed no changes. The increase in protein intake is presumed to be due to an increase in meat intake, as meat consumption and protein intake

showed a positive correlation and intake levels of nutrients that are abundant in meats, including thiamin and vitamin B6, also correlated positively with protein intake. Furthermore, there was a significant positive association between the meat consumption and the relative levels of arginine and ornithine in the blood at 3 months. Ornithine, which plays a role in urea cycle [46], is found in protein-rich foods such as meat and seafood [47]. The level of ornithine in the bloodstream can be affected by several factors, including the endogenous synthesis of ornithine from arginine facilitated by ornithine transcarbamylase, but the digestion of dietary proteins can result in an elevation of blood ornithine levels [48]. Elevated concentration of ornithine in the blood may contribute to neurological disorders caused by an excessive accumulation of ammonium [49]. In addition, high levels of ornithine has been reported to be associated with hypertension [50]. According to the 2019 Korean National Nutrition Statistics [51], the average daily meat intake of men in their 20s is 247.59 g, and that of men in their 30s is 186.64 g. Among the male subjects aged 25–35 years in this study, the ILR group's total meat intake, including processed meat, amounted to 170.6 g per day at the 3-month follow-up. In general, excessive consumption of protein, especially from meats, is not recommended, because it can cause the kidney to discharge extra nitrogen and increase the risk of osteoporosis by accelerating calcium excretion in urine [52,53]. However, the increased level of meat intake in ILR group at 3-month follow-up (170.6 g/d) was still below the national average meat intake (247.6 g/d) of Korean men in their 20s. This may be related to lower energy intake (1856.7 ± 64.7 kcal/d) and BMI (23.5 ± 0.2 kg/m$^2$) of male subjects in this study compared to the Korean men in their 20's (2020 Korea Health Statistics, Energy intake of men in their 20's: 2298.8 ± 61.1 kcal/d; BMI of men in their 20's: 25.0 ± 0.3 kg/m$^2$). Therefore, the increase in meat consumption in the ILR group at 3-month follow-up may not be a significantly adverse change. However, it will not be desirable if similar response of increased meat consumption occurs following recognition of low genetic risk in general population with fairly high meat intake.

Increased anxiety and depression levels are the most frequently reported negative psychological symptoms following genetic information recognition [20]. Although we observed no significant changes in depression and anxiety levels at any follow-up in male subjects who perceived their genetic risk to be high, significant decreases in anxiety levels at 3 months and in depression levels at both 3 and 6 months were observed in those who perceived their risk as low. The effects of genetic information disclosure on psychological factors have been investigated in several earlier studies, but in general, no strong psychological responses to DTC genetic test results have been reported. However, different results have been found depending on the severity of diseases. Studies that investigated the impact of disclosing genetic information on uncontrollable neurodegenerative disorders such as Huntington disease reported a negative psychological impact after recognizing genetic risk [54,55]. On the other hand, recognition of genetic test results regarding chronic diseases, such as cardiovascular diseases, which may be perceived to be manageable, did not result in serious increase in distress or anxiety levels [23,56,57]. However, in case of the hereditary cancers such as breast cancer and colorectal cancer, which are complex and multifactorial diseases that can be affected by environmental factors [58], varied results have been reported and the effects of disclosing genetic risk still remains inconclusive [18,20].

Nevertheless, it should be noted that the reported study results differ depending on the differences in the study design [20]. Previous studies with quantitative measures reported few impacts of disclosing genetic risk [59–61], while reviews using qualitative measurements showed negative psychological impacts of carrier testing results [62–64]. It has been reported that using standardized measures may not be sensitive enough to measure mild or moderate levels of responses [20]. Due to the complex nature of equifinality and multifinality in

psychosocial research, qualitative assessments such as condition-specific evaluations are likely to be more useful and accurate than overall psychological assessment [20]. However, qualitative studies have a limitation in that it can be subjective, and it may be difficult to reach a single conclusion when large number of subjects are included in the study. In this study, even though we measured general psychological states, significant changes in depression and anxiety levels were observed in the low genetic risk group. This can certainly be an advantage of this study, indicating that the recognizing low BMI genetic risk may induce psychological changes. There was no significant change in psychological factors in the risk perception group, but the results may need to be examined using a qualitative scale. Therefore, the lack of significance in the risk group should not be overlooked, and it is necessary to be investigated in more detail from a qualitative point of view.

In this study, only male subjects in the ILR group exhibited significant reductions in anxiety and depression levels and increases in meat intake. No significant changes were observed among female subjects. In a cross-sectional study conducted on 237 undergraduate students, it was reported that men consumed foods with high protein content more often than women [65], which is consistent with the findings of our study. We found that meat intake increased even more in male subjects who recognized their genetic risks for a high BMI and abnormal blood lipid profiles to be low. A previous study reported that women, in general, are more aware of issues regarding a healthy diet than men and tend to adopt a higher-quality diet [66]. In addition, the prevalence of obesity in Korea has been reported to be lower in women than in men during the last 10 years [67]. In this study, female subjects, even if they were aware of no genetic risk, tended to show decrease in carbohydrate intake (p = 0.067, paired t-test, Baseline; 215.2 ± 17.5 g/d, 3-month; 177.8 ± 13.8 g/d). The carbohydrate intake of the female subjects at 3-month was about 13% lower than that of the average intake of carbohydrate in females in 20's (257.6 ± 5.2 g/d, 2013–2017 Korea Health Statistics). It is suggested that women are more strict in controlling their dietary intake than men, and this may have affected sex difference in meat intake following recognition of genetic information.

A number of earlier studies have investigated the relationship between anxiety and depression and dietary intakes [6,8,68,69]. It has been proposed that greater adherence to a healthy diet, illustrated by a high score on the Alternative Healthy Eating Index-2010, is associated with a low risk of depression and anxiety [68]. Healthier diets such as the Mediterranean diet have been linked to lower levels of depressive symptoms, whereas Western diets are typically linked to a greater risk for depression [6,69]. In addition, symptoms of anxiety and depression have been associated with reduced total caloric intake but increased sugar consumption [8]. In this study, ILR group subjects showed an inverse relationship between anxiety scores and meat intake at the 3-month follow-up. In line with this finding, studies that have examined the association between a vegetarian diet and mental disorders reported that low levels of depression or anxiety are associated with a high consumption of meat [70–72]. However, other studies showed opposite findings and reported that meat avoidance is associated with better mental health [73,74]. The discrepancy between these studies may stem from the heterogeneity in assessment protocols of outcome variables [72].

This study has several strengths. First, it is a randomized controlled trial with a high follow-up rate. Second, we investigated the impact of disclosing genetic information on psychological factors and dietary intakes, as well as the relationship between dietary intake and changes in blood metabolites. Third, this study is one of the few conducted in an Asian population, which is relevant in light of the fact that dietary intake is influenced by culture and ethnicity [75–77]. This study has, however, several limitations as well. Since the study sample was limited to healthy, young, well-educated adults, it might be questionable to generalize our findings to more general populations. Although we observed statistical differences in changes in PHQ-9

and GAD-7 scores among groups, the classifications of depression and anxiety levels did not show significant deviations from normal categories throughout the study period. As a positive side note, this indicates that it is very unlikely that any of our subjects experienced unintended psychological harm. Finally, we did not observe positive changes in diet and anxiety or depression levels in subjects who perceived their genetic risk to be high. This suggests that more active interventions, going beyond just informing subjects about their genetic risks, are needed. Furthermore, intervention and advice from health professionals based on guidelines related to diet, psychology, and behaviors are needed to ensure decisive and active behavioral changes.

Taken together, the disclosure of a subject's low genetic risk for a high BMI and elevated TG and lipoprotein levels had a strong association with anxiety level and dietary intake. A decrease in anxiety and depression levels in the subjects who recognized a low genetic risk was related with an increase in dietary meat intake. This finding may provide a supportive evidence for the impact of perceived genetic risks regarding general wellness variables on changes in psychological factors and dietary patterns. Further relevant research with larger population and qualitative measures is needed to confirm the impact of disclosing genetic information on dietary intake as well as psychological factors and the association between the two.

## Supporting information

**S1 Checklist. CONSORT 2010 checklist of information to include when reporting a randomised trial\*.**
(DOC)

**S1 Table. Overview of the DTC test results provided to the study subjects.**
(DOCX)

**S2 Table. Classification of foods into 23 food groups.**
(DOCX)

**S3 Table. Generalized anxiety disorder-7 (GAD-7) and Patient Health Questionnaire-9 (PHQ-9).**
(DOCX)

**S4 Table. International physical activity questionnaire (IPAQ)-short form.**
(DOCX)

**S5 Table. Differences in food intakes among CON, ILR, and IHR groups.** [1] Change = food intake at follow-up–food intake at baseline. [2] Means with different superscripts indicate the significant differences in changes in food intakes among CON, INR, and IR groups by one-way ANOVA tests and Kruskal-Wallis tests, followed by Bonferroni-correction multiple comparison tests. The asterisks indicate a significant difference (\*: P value < 0.05) compared to food intakes from the baseline to the follow-up time point. [c] Added sugar includes sugar, fruit juice, and sugar-sweetened beverages. CON, control; ILR, Intervention-Low Risk; Intervention-High Risk.
(DOCX)

**S6 Table. Differences in anxiety and depression levels among CON, ILR, and IHR groups.** [1] Change = food intake at follow-up–food intake at baseline. [2] Means with different superscripts indicate the significant differences in changes in PHQ-9 and GAD-7 scores or categories among CON, INR, and IR groups by one-way ANOVA tests and Kruskal-Wallis tests, followed by Bonferroni-correction multiple comparison tests. The asterisks indicate a

significant difference (*: P value < 0.05) compared to PHQ-9 or GAD-7 from the baseline to the follow-up time point. CON, control; ILR, Intervention-Low Risk; Intervention-High Risk. (DOCX)

**S7 Table. Spearman's correlation between meat intake and nutrients intake.** *Significant at P<0.001. r = Spearman's correlation coefficient.
(DOCX)

**S8 Table. Multiple linear regression analysis on the association between GAD-7 or PHQ-9 scores and meat intake in total subjects.** [1] MET: metabolic equivalent task. [2] The $R^2$ values for GAD-7 scores at 3-month and 6-month were 0.072 and 0.024, respectively. The $R^2$ values for PHQ-9 scores at 3-month and 6-month were 0.050 and 0.011, respectively. ILR, Intervention-Low Risk; BMI, body mass index; PA, physical activity; GAD-7, Generalized Anxiety Disorder 7-item scale; PHQ-9, Patient Health Questionnaire 9-item scale.
(DOCX)

**S9 Table. Multiple linear analysis on the association between PHQ-9 and meat intake in the ILR group.** [1] MET: metabolic equivalent task. [2] The $R^2$ values for PHQ-9 scores at 3 months and 6 months were 0.197 and 0.055, respectively. ILR, Intervention-Low Risk; BMI, body mass index; PA, physical activity; PHQ-9, Patient Health Questionnaire 9-item scale.
(DOCX)

**S10 Table. Differences in relative levels of arginine and ornithine in the blood among CON, ILR, and IHR groups.** [1] Change = metabolite level at follow-up–metabolite level at baseline. [2] One-way ANOVA tests and Kruskal-Wallis tests were used to determine the significant differences in changes in the metabolite levels among CON, INR, and IR groups. CON, control; ILR, Intervention-Low Risk; Intervention-High Risk.
(DOCX)

**S1 File.**
(DOCX)

**S2 File.**
(DOCX)

**S3 File.**
(DOCX)

## Author Contributions

**Conceptualization:** Ga Young Lee, Jeong-Han Kim, Sung Nim Han.

**Data curation:** Ga Young Lee.

**Formal analysis:** Ga Young Lee.

**Funding acquisition:** Sung Nim Han.

**Investigation:** Ga Young Lee, Junghak Lee, Sung Nim Han.

**Methodology:** Ga Young Lee, Kyong-Mee Chung.

**Project administration:** Ga Young Lee, Sung Nim Han.

**Resources:** Ga Young Lee.

**Supervision:** Sung Nim Han.

**Visualization:** Ga Young Lee.

**Writing – original draft:** Ga Young Lee, Sung Nim Han.

**Writing – review & editing:** Ga Young Lee, Kyong-Mee Chung, Jeong-Han Kim, Sung Nim Han.

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
