## [Decision Letter · Decision Letter 0]

11 Oct 2022

PONE-D-22-19835Changes in anxiety levels and meat intake following recognition of low genetic risk for high body mass index, triglycerides, and lipoproteins: A randomized controlled trialPLOS ONE

Dear Dr. Han,

Thank you for submitting your manuscript to PLOS ONE. After careful consideration, we feel that it has merit but does not fully meet PLOS ONE’s publication criteria as it currently stands. Therefore, we invite you to submit a revised version of the manuscript that addresses the points raised during the review process.

I would like to sincerely apologize for the delay you have incurred with your submission. It has been exceptionally difficult to secure reviewers to evaluate your study. We have now received two completed reviews; the comments are available below. The reviewers have raised significant scientific concerns about the study that need to be addressed in a revision.

Please revise the manuscript to address all the reviewer's comments in a point-by-point response in order to ensure it is meeting the journal's publication criteria. Please note that the revised manuscript will need to undergo further review, we thus cannot at this point anticipate the outcome of the evaluation process.

We look forward to receiving your revised manuscript.

Kind regards,

Miquel Vall-llosera Camps

Senior Editor

PLOS ONE

Journal Requirements:

Reviewers' comments:

Reviewer's Responses to Questions

**Comments to the Author**

1. Is the manuscript technically sound, and do the data support the conclusions?

Reviewer #1: Yes

Reviewer #2: Partly

2. Has the statistical analysis been performed appropriately and rigorously? 

Reviewer #1: I Don't Know

Reviewer #2: Yes

3. Have the authors made all data underlying the findings in their manuscript fully available?

Reviewer #1: Yes

Reviewer #2: Yes

4. Is the manuscript presented in an intelligible fashion and written in standard English?

Reviewer #1: Yes

Reviewer #2: Yes

5. Review Comments to the Author

Reviewer #1: Important note: This review pertains only to ‘statistical aspects’ of the study and so ‘clinical aspects’ [like medical importance, relevance of the study, ‘clinical significance and implication(s)’ of the whole study, etc.] are to be evaluated [should be assessed] separately/independently. Further please note that any ‘statistical review’ is generally done under the assumption that (such) study specific methodological [as well as execution] issues are perfectly taken care of by the investigator(s). This review is not an exception to that and so does not cover clinical aspects {however, seldom comments are made only if those issues are intimately / scientifically related & intermingle with ‘statistical aspects’ of the study}. Agreed that ‘statistical methods’ are used as just tools here, however, they are vital part of methodology [and so should be given due importance].

COMMENTS: Your ABSTRACT is well drafted but assay type. Please note that it is preferable [refer to item 1b of CONSORT checklist 2010: Structured summary of trial design, methods, results, and conclusions] to divide the ABSTRACT with small sections like ‘Objective(s)’, ‘Methods’, ‘Results’, ‘Conclusions’, etc. which is an accepted practice of most of the good/standard journals [including this one, though ‘The PLoS One Guidelines to Authors’ did not specify an Abstract format, it is desirable]. It will definitely be more informative then, I guess, whatever the article type may be.

As you may know [it is well-known] that while reporting [findings from] ‘Clinical Trial’ one should follow CONSORT guidelines. Even important items {like How sample size was determined (Item 7a), Random Sequence generation (Item 8a), Allocation concealment (Item 9), Blinding (Item 11a)} of/in CONSORT checklist are not found [since your article type is ‘Clinical Trial’, you are supposed to cover these items in the report].

How the required minimum sample size for this study was determined is nevertheless a very-very important question for any type of study (clinical trial or else) which needs to be discussed in adequate details {including assumptions made at the time estimation, power of the study, software used, etc.}. Even how or why the allocation ratio of trial with a 2 (intervention):1 (control) subjects is decided is not given. When the allocation/assignment ratio used is different than 1:1, {which is definitely allowed but} one has to give the justification/reason (for this deviation) because then the ‘power’ [of comparison] is affected.

In section ‘5. Statistical analysis’ (line 173) it is said that “all analyses were performed using the intent-to-treat principle”. You may know well the importance & significance of ‘Per Protocol (PP)’ or Completers analyses but I request authors to refer/read section ‘5.2.3 Roles of the Different Analysis Sets’ discussed on page 24 of ‘ICH HARMONISED TRIPARTITE GUIDELINE - STATISTICAL PRINCIPLES FOR CLINICAL TRIALS, Publication-E9’.

It may also (please) be noted [in the backdrop of account given in section ‘5. Statistical analysis’, especially because you said “The differences in changes in nutrient and food intakes and psychological factors among the groups were analyzed using ANOVA for normally distributed data and Kruskal-Wallis tests for skewed data, followed by Bonferroni tests for multiple comparisons for post-hoc analysis. The changes in nutrient intakes and psychological factors were assessed using paired t tests for normally distributed data and Wilcoxon signed-rank tests for skewed data”] that for data that are in ‘ordinal’ level of measurement and not in ratio level of measurement for sure {as the score two times higher does not indicate presence of that parameter/phenomenon as double (for example, a Visual Analogue Scales VAS score or say ‘depression’ score)}. Then application of suitable non-parametric test(s) is/are indicated/advisable [even if distribution may be ‘Gaussian’ (also called ‘normal’)]. Agreed that there is/are no non-parametric test(s)/technique(s) available to be used as alternative in all situation(s) [suitable / most desired/applicable], but should be used whenever/wherever they are available.

Generally, all nutrient & food intakes and psychological factors data are in ‘ordinal’ level of measurement [definitely not in ratio level of measurement] and therefore here application of suitable non-parametric test(s) is/are indicated/advisable [even if distribution may be ‘Gaussian’ (also called ‘normal’)]. Moreover, note that ‘Multivariate linear regression’ needs continuous/ratio level data [as analyses were performed to determine the association between the psychological factors and dietary intakes].

It is appreciated that Spearman’s (and not Pearson’s) correlation coefficient was used [to assess the relationship between nutrients and meat intake] because as you know, Spearman’s correlation coefficient is non-parametric but Pearson’s correlation coefficient is parametric. However, remember that highly significant (large) value of ‘Pearson’s or Spearman’s correlation coefficient’ alone does not imply cause-effect relationship. There are other certain criteria which are to be considered before making any causal inference(s). One more important point/issue, because ‘P-value’ heavily depends on sample size, it is customary to use the absolute value of ‘Correlation coefficient’ for interpreting positive or negative correlations (and do not rely only on corresponding ‘P’-value but also consider an absolute value of ‘Correlation coefficient’).

To provide a description of baseline characteristics is entirely reasonable (since it is clearly important in assessing to whom the results of the trial can be applied), however, statistical comparison of baseline characteristics [last (6th) ‘p-value’ column in Table 1] is not desirable at all [because even if P-value turns out to be significant (while comparing baseline characteristics despite random allocation), it is, by definition, a false positive] as you then are supposed to be testing ‘randomization’ then, which in any single trial may not balance all baseline characteristics [particularly when sample sizes are small] because ‘randomization’ is a sort of ‘insurance’ and not a guarantee scheme. Authors may please refer to following articles:

References:

1. Stuart J. Pocock, et al., ‘Subgroup analysis, covariate adjustment and baseline comparisons in clinical trial reporting: current practice and problems’, Statistics in medicine, 2002; 21:2917–2930 [Particularly page 2927]

2. Harrington D, et al., ‘New guidelines for statistical reporting in the journal’, N Engl J Med 2019;381:285-6

[Important message (indirectly/ultimately indicated) from these articles: Never do any comparison with respect to ‘baseline’ characteristics {by applying statistical significance test(s)}, when allocation is done randomly].

Good that (in lines 350-354) it is clarified that “This study has several limitations as well. Since the study sample was limited to healthy, young, well-educated adults, it might be questionable to generalize our findings to more general populations. Furthermore, although we observed significant changes in depression and anxiety levels among groups, the levels stayed within the normal range throughout the study period.” however, the last part [we observed significant changes in depression and anxiety levels among groups, the levels stayed within the normal range] is not clear. What exactly do you want/wish to indicate (by saying this)?

As pointed out in ‘important note’ above “This review pertains only to ‘statistical aspects’ of the study and so ‘clinical aspects’ should be assessed separately/independently [one should carefully consider/look at the clinical implications of the study]. In my opinion, to rescue this article (which is quite possible), some amount of re-vision (re-drafting) may be needed. The respected ‘Editor’ may think of accepting only if found ‘clinical implications’ (of this study) valuable [though there are few good points of this study], I guess.

Reviewer #2: This paper examines the potential health behavior and psychological impact of providing genetic risk feedback concerning BMI, triglycerides, and lipoproteins. It’s primary strengths include the inclusion of both psychological and behavioral outcomes and the focus on Korean participants, a population that is underrepresented in this area of scholarship.

There are several areas of potential improvement for this paper:

a) This paper is predicated on the idea that genetic risk information will alter psychological states and behavioral decisions. Additional discussion must be included in the paper about the means of communication of risk, whether there were interactions researchers related to the risks, and the magnitude of the risks presented. Many of the studies referenced where participants were significantly impacted by genetic results were for variants with severe implications (BRCA1 and 2, for example), and in many cases even those serious risks did not generate major impacts. To understand the implications of this study, the authors need to provide a more detailed description of the participant experience. Some information was offered in the appended document (S1 table). Was that the extent of what the participants learned?

b) As was noted above, most research on the impact of DTC genetic testing has failed to demonstrate major shifts in psychological or behavioral states. While the paper notes some exceptions to this trend, it should provide appropriate attention to the findings that there are no major impacts (for example, in lines 65-73)

c) The major psychological outcomes were measured using established tools that evaluate overall psychological states, rather than states that are specific to the study topic. These general measures (while being widely validated) are typically not very sensitive to the types of emotional responses one would anticipate from a genetic result of this type. For this reason, many studies now use measures that focus on psychological states that are tailored to the health condition in question (for example, BRCA 1 & 2 measurement tools). The authors should discuss this tradeoff.

d) I would suggest that the authors describe the psychometrics of the measures as it relates to their findings, given that the sample size is small.

e) The authors should consider avoiding overstatement of causation between the genetic information and outcomes (as seen on line 252-4, for example), given that there are a range of other potential factors which could influence this relationship or bias responses.

f) The authors describe how the sample population is fairly different than the general population with respect to meat consumption, which is used as a key outcome. This should be discussed more extensively to provide the reader with context on why this might be the case.

g) The authors focus attention of male participants while devoting limited attention to the female participants. The lack of significant findings seems as interesting as the ones that were identified for male participants. If a demographic variable is used, the gendered implications of findings should be discussed thoroughly (note that there is a significant scholarship related to meat consumption and masculinity in the US, although I am unsure how well it translates to Korean culture).

h) A significant limitation of the study is the sample size, with the arm of the study where significant findings were observed only being composed of 16 participants.

6. PLOS authors have the option to publish the peer review history of their article (what does this mean?). If published, this will include your full peer review and any attached files.

Reviewer #1: No

Reviewer #2: No

---

## [Author Response · Author response to Decision Letter 0]

9 Nov 2022

PONE-D-22-19835

Changes in anxiety levels and meat intake following recognition of low genetic risk for high body mass index, triglycerides, and lipoproteins: A randomized controlled trial

Authors would like to thank the reviewers for constructive and valuable suggestions and comments which were very helpful in improving the manuscript. Concerns raised by reviewers have been addressed in the revised manuscript and point by point responses to comments are provided below. Substantial revision of the manuscript has been done and supporting information are provided, reflecting the reviewers’ comments.

Reviewer 1

1. This review pertains only to ‘statistical aspects’ of the study and so ‘clinical aspects’ [like medical importance, relevance of the study, ‘clinical significance and implication(s)’ of the whole study, etc.] are to be evaluated [should be assessed] separately/independently. Further please note that any ‘statistical review’ is generally done under the assumption that (such) study specific methodological [as well as execution] issues are perfectly taken care of by the investigator(s). This review is not an exception to that and so does not cover clinical aspects {however, seldom comments are made only if those issues are intimately / scientifically related & intermingle with ‘statistical aspects’ of the study}. Agreed that ‘statistical methods’ are used as just tools here, however, they are vital part of methodology [and so should be given due importance].

: Authors discussed this in the discussion part of the 'Revised Manuscript with Track Changes' as reviewer suggested (p.22, lines 376-386 and p.25, lines 458-460). 

This study shows that recognition of low genetic risk may contribute to increased meat consumption by conferring relief for their low risk. Total protein intake of male subjects in the ILR group at 3 months was 90.4 g per day, higher than the recommended dietary intake for men in their 20’s (65 g/d, 2020 Dietary Reference Intakes for Koreans). The increased level of meat intake in ILR group at 3-month follow-up (170.6 g/d) was still below the national average meat intake (247.6 g/d) of Korean men in their 20s. Therefore, the increase in meat consumption in the ILR group at 3-month follow-up may not be a significantly adverse change. However, it will not be desirable if similar response of increased meat consumption occurs following recognition of low genetic risk in general population with fairly high meat intake.

In this study, we provided genetic testing reports to subjects and did not intervene further. If more active intervention such as communicating and consulting with subjects about the test results and reinforcing healthy dietary pattern had been provided to subjects, increased meat intake would not have occurred even though a low risk was recognized. Therefore, intervention and advice from health professionals based on guidelines related to diet, psychology, and behaviors are needed to ensure decisive and active behavioral changes.

2. Your ABSTRACT is well drafted but assay type. Please note that it is preferable [refer to item 1b of CONSORT checklist 2010: Structured summary of trial design, methods, results, and conclusions] to divide the ABSTRACT with small sections like ‘Objective(s)’, ‘Methods’, ‘Results’, ‘Conclusions’, etc. which is an accepted practice of most of the good/standard journals [including this one, though ‘The PLoS One Guidelines to Authors’ did not specify an Abstract format, it is desirable]. It will definitely be more informative then, I guess, whatever the article type may be.

: Abstract has been revised to structured format by dividing to sections “Background”, “Objectives”, “Methods”, “Results”, and “Conclusions” in the revised manuscript.

3. As you may know [it is well-known] that while reporting [findings from] ‘Clinical Trial’ one should follow CONSORT guidelines. Even important items {like How sample size was determined (Item 7a), Random Sequence generation (Item 8a), Allocation concealment (Item 9), Blinding (Item 11a)} of/in CONSORT checklist are not found [since your article type is ‘Clinical Trial’, you are supposed to cover these items in the report].

: Authors explained sample size calculation (p.11, lines 195-199), random sequence generation (p.7, lines 130-131), allocation concealment (p.7, lines 131-132), and blinding (p.7, line 118 and p.8, lines 139-142) in the Method part of 'Revised Manuscript with Track Changes'.

4. How the required minimum sample size for this study was determined is nevertheless a very-very important question for any type of study (clinical trial or else) which needs to be discussed in adequate details {including assumptions made at the time estimation, power of the study, software used, etc.}. Even how or why the allocation ratio of trial with a 2 (intervention):1 (control) subjects is decided is not given. When the allocation/assignment ratio used is different than 1:1, {which is definitely allowed but} one has to give the justification/reason (for this deviation) because then the ‘power’ [of comparison] is affected.

: The sample size calculation and the rationale for the allocation ratio of trial with 2 (intervention):1 (control) are described in the Method part (p.11, lines 195-199 and p.7, 129-130) of 'Revised Manuscript with Track Changes'.

1) Sample size calculations (lines 195-199)

“We estimated the sample size by using a 1-year trial design similar to this study design [1]. Using the difference in dietary fat quality of 1.9 points between the high-apolipoprotein E genetic-risk group and low-risk group, the standard deviation of 3.6, and a power of 80% resulted in a calculated sample size of 55 per group. Considering the loss to follow-up, 65 participants were recruited for the intervention group.” 

2) Rationale for the allocation of trial with 2 (intervention):1 (control) (lines 129-130)

Recognition of their genetic information, whether an individual has genetic risks or does not have them, can induce more diverse responses in subjects compared with those not being aware of them [2]. Perception of genetic risks can 1) promote positive changes in dietary and lifestyle behaviors [3] or rather 2) induce a negative attitude such as feelings of fatalism and decrease in self-efficacy due to awareness of genetic risks [4]. On the other hand, genetic information recognition 3) may have little or no effect on health-related behaviors [5, 6]. Therefore, the ratio of the experimental group to the control group was set at 2:1 in this study, expecting that the responses after recognizing the genetic information in the experimental group would be more diverse than those of the control group not informed with their genetic information until the end of the study. This set ratio of intervention and control groups was also applied in the published paper by Ahmed El-Sohemy who investigated the relationship between genetic information recognition and behavioral changes [7].

1. Hietaranta-Luoma HL, Tahvonen R, Iso-Touru T, Puolijoki H, Hopia A. An intervention study of individual, apoE genotype-based dietary and physical-activity advice: impact on health behavior. J Nutrigenet Nutrigenomics. 2014;7(3):161-74. Epub 2015/02/28. doi: 10.1159/000371743. PubMed PMID: 25720616.

2. O'Donovan CB, Walsh MC, Gibney MJ, Brennan L, Gibney ER. Knowing your genes: does this impact behaviour change? Proc Nutr Soc. 2017;76(3):182-91. Epub 2017/01/21. doi: 10.1017/s0029665116002949. PubMed PMID: 28103960.

3. Hamburg MA, Collins FS. The path to personalized medicine. New England Journal of Medicine. 2010;363(4):301-4.

4. Bouwman LI, te Molder HF. About evidence based and beyond: a discourse-analytic study of stakeholders' talk on involvement in the early development of personalized nutrition. Health Educ Res. 2009;24(2):253-69. Epub 2008/05/24. doi: 10.1093/her/cyn016. PubMed PMID: 18499702.

5. Bloss CS, Schork NJ, Topol EJ. Effect of direct-to-consumer genomewide profiling to assess disease risk. N Engl J Med. 2011;364(6):524-34. Epub 2011/01/14. doi: 10.1056/NEJMoa1011893. PubMed PMID: 21226570; PubMed Central PMCID: PMCPMC3786730.

6. Bloss CS, Wineinger NE, Darst BF, Schork NJ, Topol EJ. Impact of direct-to-consumer genomic testing at long term follow-up. J Med Genet. 2013;50(6):393-400. Epub 2013/04/06. doi: 10.1136/jmedgenet-2012-101207. PubMed PMID: 23559530.

7. Nielsen DE, El-Sohemy A. Disclosure of genetic information and change in dietary intake: a randomized controlled trial. PLoS One. 2014;9(11):e112665. Epub 2014/11/15. doi: 10.1371/journal.pone.0112665. PubMed PMID: 25398084; PubMed Central PMCID: PMCPMC4232422 this manuscript have the following competing interests: AE-S holds shares in Nutrigenomix Inc., a genetic testing company for personalized nutrition. This does not alter the authors' adherence to PLOS ONE policies on data sharing and materials.

5. In section ‘5. Statistical analysis’ (line 173) it is said that “all analyses were performed using the intent-to-treat principle”. You may know well the importance & significance of ‘Per Protocol (PP)’ or Completers analyses but I request authors to refer/read section ‘5.2.3 Roles of the Different Analysis Sets’ discussed on page 24 of ‘ICH HARMONISED TRIPARTITE GUIDELINE - STATISTICAL PRINCIPLES FOR CLINICAL TRIALS, Publication-E9’.

: Since this study has a fairly low dropout rate (89% compliance), the results of the study were derived using the ITT principle to avoid potential bias due to exclusion of subjects. However, authors do acknowledge the importance of Per protocol analysis as well as intent-to-treat analysis, so we prepared the result of the Per Protocol analyses as a separate file, “Per Protocol analyses information”, so that readers can check the contents and compare the results with ITT analyses results.

The main findings of this study were consistent even when analyzed using Per Protocol analyses as follows. 

- A significant increase in protein intake (% energy/day) as well as meat intake (g/day) was observed in the ILR group at 3 months compared to baseline, while no significant changes were observed in the CON and IHR groups. The magnitude of change in protein intake was significantly higher in the ILR group at 3 months than in the IHR group (p=0.031 and p=0.046) (P2 Table). 

- Subjects in the ILR group showed a significant inverse association between GAD-7 scores and meat intake at 3 months (P4 Table, unadjusted model, standardized β=-0.381, p<0.05), while no significant association was observed between anxiety scores and meat intake at 6 months.

6. It may also (please) be noted [in the backdrop of account given in section ‘5. Statistical analysis’, especially because you said “The differences in changes in nutrient and food intakes and psychological factors among the groups were analyzed using ANOVA for normally distributed data and Kruskal-Wallis tests for skewed data, followed by Bonferroni tests for multiple comparisons for post-hoc analysis. The changes in nutrient intakes and psychological factors were assessed using paired t tests for normally distributed data and Wilcoxon signed-rank tests for skewed data”] that for data that are in ‘ordinal’ level of measurement and not in ratio level of measurement for sure {as the score two times higher does not indicate presence of that parameter/phenomenon as double (for example, a Visual Analogue Scales VAS score or say ‘depression’ score)}. Then application of suitable non-parametric test(s) is/are indicated/advisable [even if distribution may be ‘Gaussian’ (also called ‘normal’)]. Agreed that there is/are no non-parametric test(s)/technique(s) available to be used as alternative in all situation(s) [suitable / most desired/applicable], but should be used whenever/wherever they are available.

Generally, all nutrient & food intakes and psychological factors data are in ‘ordinal’ level of measurement [definitely not in ratio level of measurement] and therefore here application of suitable non-parametric test(s) is/are indicated/advisable [even if distribution may be ‘Gaussian’ (also called ‘normal’)]. Moreover, note that ‘Multivariate linear regression’ needs continuous/ratio level data [as analyses were performed to determine the association between the psychological factors and dietary intakes].

: Authors think that data on dietary intake and psychological factors can be used as both quantitative and qualitative variables. However, the data on nutrients and food intake and psychological factors in this study are ratio variables, not ordinal variables. Especially, GAD-7 and PHQ-9 are continuous variables that can be scored up to a total of 21 points and 27 points, respectively, depending on the degree of symptoms as described in method (p.10, lines 184-191) of 'Revised Manuscript with Track Changes'. Therefore, we used non-parametric tests such as Kruskal-Wallis tests and the Wilcoxon signed-rank tests for the data that do not follow a normal distribution.

7. It is appreciated that Spearman’s (and not Pearson’s) correlation coefficient was used [to assess the relationship between nutrients and meat intake] because as you know, Spearman’s correlation coefficient is non-parametric but Pearson’s correlation coefficient is parametric. However, remember that highly significant (large) value of ‘Pearson’s or Spearman’s correlation coefficient’ alone does not imply cause-effect relationship. There are other certain criteria which are to be considered before making any causal inference(s). One more important point/issue, because ‘P-value’ heavily depends on sample size, it is customary to use the absolute value of ‘Correlation coefficient’ for interpreting positive or negative correlations (and do not rely only on corresponding ‘P’-value but also consider an absolute value of ‘Correlation coefficient’).

: Authors do acknowledge that ‘P-value’ is not an absolute value and that ‘Correlation coefficient’ should be used to interpret the correlation of Spearman’s correlation analysis. Therefore, we provided not only the p-value, but also the correlation coefficient when presenting the results of Spearman’s correlation analysis (p.18, lines 303-305). Furthermore, as pointed out by the reviewer, special attention was paid in order to prevent correlation analysis results from being interpreted as a causal relationship when writing the Result part. The results of the regression analysis were also explained as correlations rather than causal relationships in order to avoid overstatement of causation between meat intake and anxiety.

In the Abstract and the Discussion parts, expressions which can be interpreted as a causal relationship were modified and improved to prevent misunderstanding in the 'Revised Manuscript with Track Changes'. (Abstract: p.4, line 70, Discussion: p.20, lines 334-337 and p.25, 461-465).

8. To provide a description of baseline characteristics is entirely reasonable (since it is clearly important in assessing to whom the results of the trial can be applied), however, statistical comparison of baseline characteristics [last (6th) ‘p-value’ column in Table 1] is not desirable at all [because even if P-value turns out to be significant (while comparing baseline characteristics despite random allocation), it is, by definition, a false positive] as you then are supposed to be testing ‘randomization’ then, which in any single trial may not balance all baseline characteristics [particularly when sample sizes are small] because ‘randomization’ is a sort of ‘insurance’ and not a guarantee scheme. Authors may please refer to following articles:

References:

1. Stuart J. Pocock, et al., ‘Subgroup analysis, covariate adjustment and baseline comparisons in clinical trial reporting: current practice and problems’, Statistics in medicine, 2002; 21:2917–2930 [Particularly page 2927]

2. Harrington D, et al., ‘New guidelines for statistical reporting in the journal’, N Engl J Med 2019;381:285-6

[Important message (indirectly/ultimately indicated) from these articles: Never do any comparison with respect to ‘baseline’ characteristics {by applying statistical significance test(s)}, when allocation is done randomly].

: Thank you for the detailed explanation and references on the guidelines for reporting baseline characteristics. The “p-value” column in Table 1 has been deleted in the revised manuscript.

9. Good that (in lines 350-354) it is clarified that “This study has several limitations as well. Since the study sample was limited to healthy, young, well-educated adults, it might be questionable to generalize our findings to more general populations. Furthermore, although we observed significant changes in depression and anxiety levels among groups, the levels stayed within the normal range throughout the study period.” however, the last part [we observed significant changes in depression and anxiety levels among groups, the levels stayed within the normal range] is not clear. What exactly do you want/wish to indicate (by saying this)?

: The sentence means that there was a significant difference in the GAD-7 and PHQ-9 scores among groups when the scores were considered as continuous variables; however, when the depression and anxiety were compared as stages of symptoms, which are categorical variables, all of the study subjects were within the normal range of depression and anxiety levels. We modified the sentence to further clarify the meaning in the 'Revised Manuscript with Track Changes' (p.25, lines 449-451). 

“Furthermore, although we observed significant changes in depression and anxiety levels among groups, the levels stayed within the normal range throughout the study period.”

“Although we observed statistical differences in changes in PHQ-9 and GAD-7 scores among groups, the categories of both depression and anxiety symptoms were maintained within the normal stage throughout the study period.”

As pointed out in ‘important note’ above “This review pertains only to ‘statistical aspects’ of the study and so ‘clinical aspects’ should be assessed separately/independently [one should carefully consider/look at the clinical implications of the study]. In my opinion, to rescue this article (which is quite possible), some amount of re-vision (re-drafting) may be needed. The respected ‘Editor’ may think of accepting only if found ‘clinical implications’ (of this study) valuable [though there are few good points of this study], I guess.

 

Reviewer 2 

This paper examines the potential health behavior and psychological impact of providing genetic risk feedback concerning BMI, triglycerides, and lipoproteins. It’s primary strengths include the inclusion of both psychological and behavioral outcomes and the focus on Korean participants, a population that is underrepresented in this area of scholarship. 

There are several areas of potential improvement for this paper:

1. This paper is predicated on the idea that genetic risk information will alter psychological states and behavioral decisions. Additional discussion must be included in the paper about the means of communication of risk, whether there were interactions researchers related to the risks, and the magnitude of the risks presented. Many of the studies referenced where participants were significantly impacted by genetic results were for variants with severe implications (BRCA1 and 2, for example), and in many cases even those serious risks did not generate major impacts. To understand the implications of this study, the authors need to provide a more detailed description of the participant experience. Some information was offered in the appended document (S1 table). Was that the extent of what the participants learned?

: Authors provided description on the details regarding how intervention was proceeded in the method part of the 'Revised Manuscript with Track Changes' (p.8, lines 147-152). 

In this study, providing the genetic risk results to the subjects was the sole intervention. Subjects in the intervention group were provided with the DTC genetic test reports in a written form at the beginning of the intervention, and no further interactions such as explanation of the test result or interaction with researchers were performed. Information included in the DTC genetic test report was their genotype of each SNPS of the genes regarding BMI, TG, LDL- and HDL-cholesterol levels, blood pressure, and caffeine metabolism, magnitude of the risk (“Good” for not having genotype at risk, “Borderline risk” for carrying one genotype at risk, “Caution” for having two genotypes at risk), and descriptions of and recommendations regarding the risk alleles (described in the S1 table). No further information or communication with researchers were provided as described in p.8, line 139. Among the genetic items that were provided in the DTC test report, only genetic items regarding BMI, TG, LDL- and HDL-cholesterol levels came with recommendations related to diet and exercise which can affect subject’s behavior. Therefore, in this study, genetic items related to BMI, TG, LDL-, and HDL-cholesterol levels were selected as genetic information to be provided to the subjects. 

This study is different from previous studies which provided genetic information with severe implications. Authors thought that people who already have diseases such as obesity, diabetes, or cancers were likely to make some extent of life-style modifications. Therefore, we have decided to provide DTC genetic information on BMI, TG, and lipoproteins to investigate changes in health-related behaviors for disease prevention and health management upon recognition of genetic risk, as described in p.21, lines 357-361.

2. As was noted above, most research on the impact of DTC genetic testing has failed to demonstrate major shifts in psychological or behavioral states. While the paper notes some exceptions to this trend, it should provide appropriate attention to the findings that there are no major impacts (for example, in lines 65-73)

: In authors’ opinion, it is difficult to draw a decisive conclusion regarding the impact of DTC genetic testing on anxiety and depression or behavioral changes because there are only few studies available in which the impact of DTC genetic information disclosure on psychological factors has been investigated. Furthermore, studies have reported conflicting results. Various methodologies used in those studies may have induced heterogeneous results among studies. 

In this study, we found different responses between those who perceived high genetic risk and those who perceived low genetic risk. Although there were no significant changes in diet and psychological factors in the subjects who perceived high genetic risk, we found that the level of depression and anxiety levels decreased in the subjects who perceived low genetic risk. Previous studies referenced in the discussion part (p.22, lines 395-399) also reported that there was no change in depression and anxiety levels in people with BRACA1 mutation, but a decrease in psychological distress was observed in those without the mutation [8, 9]. A reduced psychological distress will improve quality of life and will have beneficial effect on chronic diseases affected by stress; however, a decrease in the levels of depression and anxiety due to relief does not necessarily result in desirable changes in terms of behavior. In this study, an increase in meat consumption was observed in subjects who recognized that their genetic risk was low.

8. Butow PN, Lobb EA, Meiser B, Barratt A, Tucker KM. Psychological outcomes and risk perception after genetic testing and counselling in breast cancer: a systematic review. Med J Aust. 2003;178(2):77-81. doi: 10.5694/j.1326-5377.2003.tb05069.x. PubMed PMID: 12526728.

9. Kinney AY, Bloor LE, Mandal D, Simonsen SE, Baty BJ, Holubkov R, et al. The impact of receiving genetic test results on general and cancer-specific psychologic distress among members of an African-American kindred with a BRCA1 mutation. Cancer. 2005;104(11):2508-16. doi: 10.1002/cncr.21479. PubMed PMID: 16222692.

3. The major psychological outcomes were measured using established tools that evaluate overall psychological states, rather than states that are specific to the study topic. These general measures (while being widely validated) are typically not very sensitive to the types of emotional responses one would anticipate from a genetic result of this type. For this reason, many studies now use measures that focus on psychological states that are tailored to the health condition in question (for example, BRCA 1 & 2 measurement tools). The authors should discuss this tradeoff.

: It seems that the expression “psychological status” did not accurately reflect the authors’ intention. In this study, we focused mainly on depression and anxiety, the most well-known psychological status. Therefore, we included “depression” in the title to reflect both anxiety and depression levels as psychological factors and changed the text “psychological status” within the manuscript to “anxiety or depressions levels” in order to be focused and be specific.

4. I would suggest that the authors describe the psychometrics of the measures as it relates to their findings, given that the sample size is small.

: The internal consistency expressed as Cronbach's α in the Korean version of validation study of GAD-7 was 0.92 [10], and the Cronbach's α in this study was 0.91. Regarding PHQ-9, the internal consistency reported in the Korean version validation study was 0.95 [11], and this study had a Cronbach's α of 0.88. Therefore, the psychometrics of the measures used in this study has been verified to be appropriate, and the authors described in the 'Revised Manuscript with Track Changes' (p.10, lines 186-188 and lines 191-192).

10. Seo JG, Park SP. Validation of the Generalized Anxiety Disorder-7 (GAD-7) and GAD-2 in patients with migraine. J Headache Pain. 2015;16:97. Epub 20151123. doi: 10.1186/s10194-015-0583-8. PubMed PMID: 26596588; PubMed Central PMCID: PMCPMC4656257.

11. An JY, Seo ER, Lim KH, Shin JH, Kim JB. Standardization of the Korean version of screening tool for depression (Patient Health Questionnaire-9, PHQ-9). J Korean Soc Biol Ther Psychiatry. 2013;19(1):47-56.

5. The authors should consider avoiding overstatement of causation between the genetic information and outcomes (as seen on line 252-4, for example), given that there are a range of other potential factors which could influence this relationship or bias responses.

: Authors discussed about this and modified expressions in the Abstract (p.4, line 70) and Discussion parts (p.20, lines 334-337 and p.25, lines 461-465) of the 'Revised Manuscript with Track Changes' to avoid interpretation as causal relationships.

6. The authors describe how the sample population is fairly different than the general population with respect to meat consumption, which is used as a key outcome. This should be discussed more extensively to provide the reader with context on why this might be the case.

: Authors provided comparison of meat consumption and protein intake by subjects in the study with the 2020 Dietary Reference Intakes for Koreans (KDRIs) and with average intake by general population, and discussed the differences in the Discussion part of the 'Revised Manuscript with Track Changes' (p.22, line 376-383).

 “Total protein intake of male subjects in the ILR group at 3 months was 90.4 g per day, higher than the recommended dietary intake for men in their 20’s (65 g/d, 2020 Dietary Reference Intakes for Koreans). The increased level of meat intake in ILR group at 3-month follow-up (170.6 g/d) was still below the national average meat intake (247.6 g/d) of Korean men in their 20s. This may be related to lower energy intake (1856.7 ± 64.7 kcal/d) and BMI (23.5 ± 0.2 kg/m2) of male subjects in this study compared to the Korean men in their 20’s (2020 Korea Health Statistics, Energy intake of men in their 20’s: 2298.8 ± 61.1 kcal/d; BMI of men in their 20’s: 25.0 ± 0.3 kg/m2). “ 

7. The authors focus attention of male participants while devoting limited attention to the female participants. The lack of significant findings seems as interesting as the ones that were identified for male participants. If a demographic variable is used, the gendered implications of findings should be discussed thoroughly (note that there is a significant scholarship related to meat consumption and masculinity in the US, although I am unsure how well it translates to Korean culture).

: Authors do agree that it is important to point out gender difference in meat intake, and gendered implications of the findings should be discussed. We discussed these in the 'Revised Manuscript with Track Changes' (p.23, line 415-423). Below is the added discussion in the revised manuscript.

“A previous study reported that, in general, women are more aware of issues regarding a healthy diet than men and tend to adopt a higher-quality diet [12]. In addition, the prevalence of obesity in Korea has been reported to be lower in women than in men during the last 10 years [13]. In this study, female subjects, even if they were aware of no genetic risk, tended to show decrease in carbohydrate intake (p=0.067, paired t-test, Baseline; 215.2 ± 17.5 g/day, 3-month; 177.8 ± 13.8 g/day). The carbohydrate intake of the female subjects at 3-month was about 13 %lower than that of the average intake of carbohydrate in females in 20's (257.6 ± 5.2 g/day, 2013-2017 Korea Health Statistics). It is suggested that women are stricter in controlling their dietary intake than men, and this may have affected gender difference in meat intake following recognition of genetic information.”

When looking at the relationship between meat intake and masculinity, much has been reported about the meat intake and masculinity in US and Europe. In Western population, women are two times more likely to be a vegan than men [14]. However, there is lack of research in Korea regarding the relationship between meat intake and masculinity. 

12. Leblanc V, Bégin C, Corneau L, Dodin S, Lemieux S. Gender differences in dietary intakes: what is the contribution of motivational variables? J Hum Nutr Diet. 2015;28(1):37-46. Epub 20140214. doi: 10.1111/jhn.12213. PubMed PMID: 24527882.

13. Yoo S, Cho HJ, Khang YH. General and abdominal obesity in South Korea, 1998-2007: gender and socioeconomic differences. Prev Med. 2010;51(6):460-5. Epub 20101016. doi: 10.1016/j.ypmed.2010.10.004. PubMed PMID: 20955726.

14. Ruby MB. Vegetarianism. A blossoming field of study. Appetite. 2012;58(1):141-50. Epub 20111004. doi: 10.1016/j.appet.2011.09.019. PubMed PMID: 22001025.

8. A significant limitation of the study is the sample size, with the arm of the study where significant findings were observed only being composed of 16 participants.

: As we presented in the Table 3, the significant negative correlation between the GAD-7 score and meat intake was verified in the results of the regression analysis performed within the ILR group with gender as a confounding variable, so it can be said that the significant findings were observed in 32 participants. 

Furthermore, we provided sample size calculation in the Method part of the 'Revised Manuscript with Track Changes' (p.11, line 195-199). In this study, the subjects included in the experimental group (n=65) were further subdivided into the intervention-low risk group (ILR, n=32) and the intervention-high risk group (IHR, n=33) depending on the degree of genetic risk, so the number of subjects included in each intervention group was similar to the control group (CON, 35). 

However, authors do acknowledge that sample size of the study participants is not large enough to generalize the findings of the study. Despite this limitation, authors believe that it is worth discussing the relationship between genetic information recognition and meat consumption from a gender-specific perspective. What makes this study more meaningful is that there are few studies which have observed the effect of genetic information recognition on changes in psychological factors and changes in dietary intakes together, and it is hard to find studies conducted especially on East Asians. Although further studies with larger population size are needed to confirm and elaborate the findings from this study, as provided as suggestions in p.25, line 467-469, authors believe that this study can contribute to accumulation of data for developing a scientific framework for the implementation of personalized nutrition from a long-term perspective.

---

## [Decision Letter · Decision Letter 1]

22 Dec 2022

PONE-D-22-19835R1Changes in anxiety and depression levels and meat intake following recognition of low genetic risk for high body mass index, triglycerides, and lipoproteins: A randomized controlled trialPLOS ONE

Dear Dr. Han,

Thank you for submitting your manuscript to PLOS ONE. After careful consideration, we feel that it has merit but does not fully meet PLOS ONE’s publication criteria as it currently stands. Therefore, we invite you to submit a revised version of the manuscript that addresses the points raised during the review process.

The manuscript has been evaluated by two reviewers, and their comments are available below.

The reviewers have raised a number of remaining concerns. They feel the manuscript should give further context, include additional details in the discussion, and further explain the rationale for the trial allocation ratio of 2 (intervention):1 (control).

Could you please carefully revise the manuscript to address all comments raised?

We look forward to receiving your revised manuscript.

Kind regards,

Alice Coles-Aldridge

Editorial Office

PLOS ONE

Reviewers' comments:

Reviewer's Responses to Questions

**Comments to the Author**

1. If the authors have adequately addressed your comments raised in a previous round of review and you feel that this manuscript is now acceptable for publication, you may indicate that here to bypass the “Comments to the Author” section, enter your conflict of interest statement in the “Confidential to Editor” section, and submit your "Accept" recommendation.

Reviewer #1: All comments have been addressed

Reviewer #2: (No Response)

2. Is the manuscript technically sound, and do the data support the conclusions?

Reviewer #1: (No Response)

Reviewer #2: Yes

3. Has the statistical analysis been performed appropriately and rigorously? 

Reviewer #1: (No Response)

Reviewer #2: Yes

4. Have the authors made all data underlying the findings in their manuscript fully available?

Reviewer #1: (No Response)

Reviewer #2: Yes

5. Is the manuscript presented in an intelligible fashion and written in standard English?

Reviewer #1: (No Response)

Reviewer #2: Yes

6. Review Comments to the Author

Reviewer #1: COMMENTS: Since few {not all correctly} of the comments made on earlier draft [but frankly speaking, I am not very much convinced (or happy) for reasons given or arguments made {particularly the rationale for the allocation ratio of trial with 2 (intervention):1 (control)}, I feel, ‘let the respected editor decide the future course’. Note that ‘Response to Reviewers’ are not given separately as required [making it difficult read]. I do not have any specific recommendation [though only as system requirement, I choose minor revision].

Reviewer #2: I appreciate the authors thoughtful response to the previously offered comments. In some cases the comments were fully answered, and they have been removed from this response. In other cases I have offered an elaboration.

REVIEWERS ORIGINAL POINT #2. As was noted above, most research on the impact of DTC genetic testing has failed to demonstrate major shifts in psychological or behavioral states. While the paper notes some exceptions to this trend, it should provide appropriate attention to the findings that there are no major impacts (for example, in lines 65-73)

AUTHORS RESPONSE TO POINT #2: In authors’ opinion, it is difficult to draw a decisive conclusion regarding the impact of DTC genetic testing on anxiety and depression or behavioral changes because there are only few studies available in which the impact of DTC genetic information disclosure on psychological factors has been investigated. Furthermore, studies have reported conflicting results. Various methodologies used in those studies may have induced heterogeneous results among studies.

In this study, we found different responses between those who perceived high genetic risk and those who perceived low genetic risk. Although there were no significant changes in diet and psychological factors in the subjects who perceived high genetic risk, we found that the level of depression and anxiety levels decreased in the subjects who perceived low genetic risk. Previous studies referenced in the discussion part (p.22, lines 395-399) also reported that there was no change in depression and anxiety levels in people with BRACA1 mutation, but a decrease in psychological distress was observed in those without the mutation [8, 9]. A reduced psychological distress will improve quality of life and will have beneficial effect on chronic diseases affected by stress; however, a decrease in the levels of depression and anxiety due to relief does not necessarily result in desirable changes in terms of behavior. In this study, an increase in meat consumption was observed in subjects who recognized that their genetic risk was low.

8.Butow PN, Lobb EA, Meiser B, Barratt A, Tucker KM. Psychological outcomes and risk perception after genetic testing and counselling in breast cancer: a systematic review. Med J Aust. 2003;178(2):77-81. doi: 10.5694/j.1326-5377.2003.tb05069.x.PubMed PMID: 12526728.

9.Kinney AY, Bloor LE, Mandal D, Simonsen SE, Baty BJ, Holubkov R, et al. The impact of receiving genetic test results on general and cancer-specific psychologic distress among members of an African-American kindred with a BRCA1 mutation. Cancer. 2005;104(11):2508-16. doi: 10.1002/cncr.21479. PubMed PMID: 16222692.

REVIEWERS RESPONSE TO AUTHORS ON POINT #2: There may be some miscommunication on this point. Given the topic that is being explored, it is crucial to contextualize it in the existing literature which has not generally indicated a strong psychosocial or behavioral response to DTC genetic results. This makes the findings of this study unusual, although not entirely inconsistent with occasional impacts that are observed. It is important to distinguish between responses from severe risks and minor risks as are conveyed in these DTC findings that were communicated to participants, and the literature review should be careful to not conflate the two. References #8 and 9 cited above are interesting, but largely obsolete, being over 15 years old. DTC testing did not become widely available until after 2007, which was after the publication of these papers. There has been widespread and extensive research on the topics examined by this scholarship that need to be acknowledged and integrated. For example, please see the review of 37 papers on the topic synthesized by Covolo et al, 2015.

Furthermore, for a review that explores a range of systematic reviews on this topic, consider examining Wade (2019).

Wade, Christopher H., “ What Is the Psychosocial Impact of Providing Genetic and Genomic Health Information to Individuals? An Overview of Systematic Reviews,” Looking for the Psychosocial Impacts of Genomic Information, special report, Hastings Center Report 49, no. 3 (2019): S88– S96. DOI: 10.1002/hast.1021

It is likely that there are more recent reviews which could be used; and the authors are encouraged to look for them. The context of this scholarship, and the way the this study is situated within that field of study, should be developed further in both the introduction and the discussion sections.

REVIEWERS ORIGINAL POINT # 3. The major psychological outcomes were measured using established tools that evaluate overall psychological states, rather than states that are specific to the study topic. These general measures (while being widely validated) are typically not very sensitive to the types of emotional responses one would anticipate from a genetic result of this type. For this reason, many studies now use measures that focus on psychological states that are tailored to the health condition in question (for example, BRCA 1 & 2 measurement tools). The authors should discuss this tradeoff.

AUTHORS RESPONSE TO POINT #3: It seems that the expression “psychological status” did not accurately reflect the authors’ intention. In this study, we focused mainly on depression and anxiety, the most well-known psychological status. Therefore, we included “depression” in the title to reflect both anxiety and depression levels as psychological factors and changed the text “psychological status” within the manuscript to “anxiety or depressions levels” in order to be focused and be specific.

REVIEWERS RESPONSE TO AUTHORS ON POINT #3: This comment was more specifically focused on the measures used to evaluate psychological states. The measures used were standardized measures for global states, which tend to be insensitive to the mild/moderate responses typically observed for DTC genetic risk findings. Test-specific measures tend to perform better. To examine this issue further, please refer to the reviews mentioned above. It would be beneficial to comment on the measures used, the fact that they measure general states, and the implications for observing a significant difference in the discussion, particularly in light of the findings from other studies in this field.

GENERAL ADDED COMMENT: The authors may wish to add a concluding paragraph.

7. PLOS authors have the option to publish the peer review history of their article (what does this mean?). If published, this will include your full peer review and any attached files.

Reviewer #1: No

Reviewer #2: No

---

## [Author Response · Author response to Decision Letter 1]

8 Jan 2023

Response to reviewers’ comments 

PONE-D-22-19835R1

Changes in anxiety and depression levels and meat intake following recognition of low genetic risk for high body mass index, triglycerides, and lipoproteins: A randomized controlled trial

PLOS ONE

Authors would like to thank the reviewers for kind explanations and valuable suggestions and comments that are necessary to improve this paper. Authors have acknowledged the concerns raised by reviewers and provided the response to reviewers’ comments and the revised version of the manuscript.

Reviewer #1

COMMENTS: Since few [1] of the comments made on earlier draft [but frankly speaking, I am not very much convinced (or happy) for reasons given or arguments made {particularly the rationale for the allocation ratio of trial with 2 (intervention):1 (control)}, I feel, ‘let the respected editor decide the future course’. Note that ‘Response to Reviewers’ are not given separately as required [making it difficult read]. I do not have any specific recommendation [though only as system requirement, I choose minor revision].

: Authors would like to apologize that the response was not clear enough and that the reviewer was not convinced with the rationale for the allocation. We have provided further explanation and reference for this. We applied a 2:1 ratio of subjects in the intervention group and the control group because the intervention group consisted of those with either “risk” or “no-risk” genotypes of genes in the genetic testing results [1]. This is provided as the rationale for the allocation ratio of trial with 2 (intervention): 1 (control) in the method part of the ‘Revised manuscript with track changes’ (p.8, lines 124-126). 

1. Nielsen DE, El-Sohemy A. Disclosure of genetic information and change in dietary intake: a randomized controlled trial. PLoS One. 2014;9(11):e112665. Epub 2014/11/15. doi: 10.1371/journal.pone.0112665. PubMed PMID: 25398084; PubMed Central PMCID: PMCPMC4232422 this manuscript has the following competing interests: AE-S holds shares in Nutrigenomix Inc., a genetic testing company for personalized nutrition. This does not alter the authors' adherence to PLOS ONE policies on data sharing and materials.

Reviewer #2 

I appreciate the authors thoughtful response to the previously offered comments. In some cases the comments were fully answered, and they have been removed from this response. In other cases I have offered an elaboration.

REVIEWERS RESPONSE TO AUTHORS ON POINT #2: There may be some miscommunication on this point. Given the topic that is being explored, it is crucial to contextualize it in the existing literature which has not generally indicated a strong psychosocial or behavioral response to DTC genetic results. This makes the findings of this study unusual, although not entirely inconsistent with occasional impacts that are observed. It is important to distinguish between responses from severe risks and minor risks as are conveyed in these DTC findings that were communicated to participants, and the literature review should be careful to not conflate the two. References #8 and 9 cited above are interesting, but largely obsolete, being over 15 years old. DTC testing did not become widely available until after 2007, which was after the publication of these papers. There has been widespread and extensive research on the topics examined by this scholarship that need to be acknowledged and integrated. For example, please see the review of 37 papers on the topic synthesized by Covolo et al, 2015.

Furthermore, for a review that explores a range of systematic reviews on this topic, consider examining Wade (2019).

Wade, Christopher H., “ What Is the Psychosocial Impact of Providing Genetic and Genomic Health Information to Individuals? An Overview of Systematic Reviews,” Looking for the Psychosocial Impacts of Genomic Information, special report, Hastings Center Report 49, no. 3 (2019): S88– S96. DOI: 10.1002/hast.1021

It is likely that there are more recent reviews which could be used; and the authors are encouraged to look for them. The context of this scholarship, and the way this study is situated within that field of study, should be developed further in both the introduction and the discussion sections.

: Thank you very much for the detailed explanation and recommending reference of the recent review. The authors reviewed previous studies, focusing on the latest researches, and summarized the overall flow of the studies and the results according to the severity of the diseases in the discussion part of the ‘Revised manuscript with track changes’ (p.23-24, lines 384-396). Furthermore, we provided a revised introduction part reflecting the above point of view (p.5-6, lines 77-85). We have cited the more recent references in the revised manuscript. 

REVIEWERS RESPONSE TO AUTHORS ON POINT #3: This comment was more specifically focused on the measures used to evaluate psychological states. The measures used were standardized measures for global states, which tend to be insensitive to the mild/moderate responses typically observed for DTC genetic risk findings. Test-specific measures tend to perform better. To examine this issue further, please refer to the reviews mentioned above. It would be beneficial to comment on the measures used, the fact that they measure general states, and the implications for observing a significant difference in the discussion, particularly in light of the findings from other studies in this field.

GENERAL ADDED COMMENT: The authors may wish to add a concluding paragraph.

: Authors acknowledged the need to comment on the measures we used and to discuss implications for the results of this study by mentioning the results of other studies that used condition-specific assessments. Therefore, we discussed this in the discussion part (p.24, lines 397-413) of the ‘Revised Manuscript with Track Changes’ and the modified introduction (p.5, lines 80-82) and conclusion (p.27, lines 477-481).

---

## [Decision Letter · Decision Letter 2]

10 Jul 2023

PONE-D-22-19835R2Changes in anxiety and depression levels and meat intake following recognition of low genetic risk for high body mass index, triglycerides, and lipoproteins: A randomized controlled trialPLOS ONE

Dear Dr. Han,

Thank you for submitting your manuscript to PLOS ONE. After careful consideration, we feel that it has merit but does not fully meet PLOS ONE’s publication criteria as it currently stands. Therefore, we invite you to submit a revised version of the manuscript that addresses the points raised during the review process.

We look forward to receiving your revised manuscript.

Kind regards,

Reindolf Anokye

Academic Editor

PLOS ONE

Journal Requirements:

Reviewers' comments:

Reviewer's Responses to Questions

**Comments to the Author**

1. If the authors have adequately addressed your comments raised in a previous round of review and you feel that this manuscript is now acceptable for publication, you may indicate that here to bypass the “Comments to the Author” section, enter your conflict of interest statement in the “Confidential to Editor” section, and submit your "Accept" recommendation.

Reviewer #1: (No Response)

Reviewer #3: (No Response)

2. Is the manuscript technically sound, and do the data support the conclusions?

Reviewer #1: (No Response)

Reviewer #3: Yes

3. Has the statistical analysis been performed appropriately and rigorously? 

Reviewer #1: (No Response)

Reviewer #3: I Don't Know

4. Have the authors made all data underlying the findings in their manuscript fully available?

Reviewer #1: (No Response)

Reviewer #3: Yes

5. Is the manuscript presented in an intelligible fashion and written in standard English?

Reviewer #1: (No Response)

Reviewer #3: Yes

6. Review Comments to the Author

Reviewer #1: COMMENTS: As earlier, I do not have any specific recommendation [though only as system requirement I choose minor revision]. Let the respected editor decide the future course. Nevertheless, I do not have any reservations.

Reviewer #3: What is the necessity of doing this study? What is the practical purpose of this study? What can the results of this study help people?

Doesn't people's awareness of genetic predisposition to obesity lead to a lack of motivation to improve their lifestyle? Considering that lifestyle plays the most important role in the occurrence of obesity or dyslipidemia, what is the importance of knowing genetic predisposition? it was also seen in your results that low-risk participants increased meat consumption. Maybe because they didn't see meat and saturated fatty acids as a danger to their health

It seems that you also gave advice to high-risk people. Doesn't this affect their dietary intake?

Considering that depression can be a hidden disease, how did you include this in the inclusion criteria?

What is your suggestion for future studies in this field?

Didn't you measure lipoprotein and TG at baseline? As, who have dyslipidemia usually have an unhealthy lifestyle, which can affect the results.

How did you assess physical activity? Please describe in the method section

7. PLOS authors have the option to publish the peer review history of their article (what does this mean?). If published, this will include your full peer review and any attached files.

Reviewer #1: No

Reviewer #3: No

---

## [Author Response · Author response to Decision Letter 2]

24 Jul 2023

PONE-D-22-19835R2

Changes in anxiety and depression levels and meat intake following recognition of low genetic risk for high body mass index, triglycerides, and lipoproteins: A randomized controlled trial

PLOS ONE

Authors would like to thank the reviewers for constructive and valuable suggestions and comments. Concerns raised by reviewers are addressed in the 3rd revision of the manuscript, and point by point responses to comments are provided. Furthermore, we included blood metabolite analysis results that became available after initial submission of the manuscript in order explain the implications of the increased meat intake in the ILR group. (lines 49-50, 59-61, 195-209, 345-359, 395-403, 484-486. By addition of the metabolite profile results, we aimed to present the potential issues associated with increased meat consumption. Authors who are responsible for the metabolite-related results are included as co-authors in the revised manuscript.

Review Comments to the Author

Reviewer #1: COMMENTS: As earlier, I do not have any specific recommendation [though only as system requirement I choose minor revision]. Let the respected editor decide the future course. Nevertheless, I do not have any reservations.

Reviewer #3: What is the necessity of doing this study? What is the practical purpose of this study? What can the results of this study help people?

Doesn't people's awareness of genetic predisposition to obesity lead to a lack of motivation to improve their lifestyle? Considering that lifestyle plays the most important role in the occurrence of obesity or dyslipidemia, what is the importance of knowing genetic predisposition? it was also seen in your results that low-risk participants increased meat consumption. Maybe because they didn't see meat and saturated fatty acids as a danger to their health

It seems that you also gave advice to high-risk people. Doesn't this affect their dietary intake?

Considering that depression can be a hidden disease, how did you include this in the inclusion criteria?

What is your suggestion for future studies in this field?

Didn't you measure lipoprotein and TG at baseline? As, who have dyslipidemia usually have an unhealthy lifestyle, which can affect the results.

How did you assess physical activity? Please describe in the method section

① What is the necessity of doing this study? What is the practical purpose of this study? What can the results of this study help people?

: The use of DTC genetic tests has been increasing as a tool for precision nutrition in maintaining and managing an individual's optimal health condition. Previous studies have mostly focused on exploring the impact disclosing DTC genetic information related to diseases on behavior changes. However, the authors believed it was essential to investigate the effect of recognizing genetic information related to wellness areas, such as BMI and blood lipid profile. This is important because adopting positive health-related behavior changes, even before the onset of diseases, can result in significant benefits for health management. Moreover, given the nature of DTC genetic tests, customers often receive genetic test results directly without any guidance or intervention from nutritionist or healthcare experts. Therefore, we wanted to examine whether consumers would initiate behavioral changes based solely on the genetic test results provided to them, and we aimed to investigate the effect of recognizing genetic information regarding BMI, TG, and lipoproteins on health-related behavior changes in this study. 

The study findings revealed that individuals in the ILR group, who perceived their genetic risk to be low, exhibited an increase in meat consumption. The results of this study highlight the fact that simply providing people with the results of their genetic information is insufficient for improving health-related behaviors. Moreover, results demonstrate that genetic information can lead to unhealthy behavior changes when people perceive their risk to be low. Ultimately, this study emphasizes the significance of offering interventions and advice from healthcare professionals based on guidelines related to diet, psychology, and behavior. Such interventions can facilitate effective and proactive behavioral changes, as described in the manuscript (lines 497-500).

② Doesn't people's awareness of genetic predisposition to obesity lead to a lack of motivation to improve their lifestyle? Considering that lifestyle plays the most important role in the occurrence of obesity or dyslipidemia, what is the importance of knowing genetic predisposition? it was also seen in your results that low-risk participants increased meat consumption. Maybe because they didn't see meat and saturated fatty acids as a danger to their health. It seems that you also gave advice to high-risk people. Doesn't this affect their dietary intake?

: In addition to lifestyle factors, genetic factors have a significant impact on the development of obesity and dyslipidemia. While being aware of genetic information related to severe and unmodifiable genetic disorders may discourage individuals from making behavioral changes, individuals may adopt changes in health-related behaviors when modifiable health-related genetic information is provided. However, the results of this study emphasize that simply being aware of the genetic information is insufficient to motivate people to improve their health-related behaviors. It highlights the necessity of active involvement by experts. 

In this study, the DTC genetic test reports included specific recommendations for individuals with risk alleles (e.g. recommendation of low-fat diet for people with BMI-related risk alleles), whereas no recommendations regarding health-related behaviors were given to subjects without risk alleles. As a result, subjects in the ILR group were less attentive to their health behaviors due to their low genetic risk perception, which may have led to an increase in their meat intake. On the other hand, the fact that the dietary intake in the IHR group did not worsen, despite not showing improvement, may be may be attributed to the recommendations provided to the high-risk subjects.

③ Considering that depression can be a hidden disease, how did you include this in the inclusion criteria?

: The study participants had an average PHQ-9 score of 5.1 ± 4.6, placing them within the range of mild depression based on the PHQ-9 test criteria (1-4: normal, 5-9: mild depression, 10-19: moderate depression, 20 and above: severe depression). However, it is important to note that these individuals did not have a medical history of being diagnosed with depression by healthcare professionals, and the recruitment process excluded individuals with symptoms of depression or those taking medication for depression. As a result, the study participants did not exhibit a clinically significant level of depression.

Furthermore, while depression is often associated with symptoms of anxiety, the average GAD-7 score of the study participants at baseline was 3.7 ± 3.5, which falls within the normal range on the GAD-7 test (0-4: normal, 5-9: mild anxiety, 10-14: moderate anxiety, 15-21: severe anxiety). These indicates the absence of anxiety symptoms among the study participants.

④ What is your suggestion for future studies in this field?

: The results from this study suggest that simply providing genetic test results is not enough to bring significant changes in health-related behaviors for majority of people. Therefore, if further research is conducted, it is suggested to investigate the potential differences in the impact of personalized recommendations based on the genetic information related to nutrient metabolism and metabolic diseases compared to more general dietary advice for managing metabolic conditions. 

⑤ Didn't you measure lipoprotein and TG at baseline? As, who have dyslipidemia usually have an unhealthy lifestyle, which can affect the results.

: We appreciate the valuable suggestions. In this study, we evaluated the levels of total cholesterol, LDL- and HDL-cholesterol, and TG at the baseline. The results showed that there were no significant differences among the CON, ILR, and IHR groups (Total cholesterol: p=0.158; LDL-cholesterol: p=0.803; HDL-cholesterol: p=0.453; TG: p=0.393, determined by One-way ANOVA test). Moreover, the average baseline levels of total cholesterol and triglyceride were within the normal range (Total cholesterol < 100 mg/dL; triglyceride < 150 mg/dL).

The revised manuscript now includes the average baseline measurements of total cholesterol, LDL- and HDL-cholesterol, and TG in Table 1.

⑥ How did you assess physical activity? Please describe in the method section

: In response to reviewer’s feedback, we have included the details of the procedure for measuring physical activity in the revised manuscript's methodology section (lines 185-193).

---

## [Editor Report · Decision Letter 3]

22 Aug 2023

Changes in anxiety and depression levels and meat intake following recognition of low genetic risk for high body mass index, triglycerides, and lipoproteins: A randomized controlled trial

PONE-D-22-19835R3

Dear Dr. Han,

We’re pleased to inform you that your manuscript has been judged scientifically suitable for publication and will be formally accepted for publication once it meets all outstanding technical requirements.

Kind regards,

Reindolf Anokye

Academic Editor

PLOS ONE

Additional Editor Comments (optional):

Ensure that all grammatical errors are corrected.

Please ensure that all references cited in the manuscript are included in the list of references, and vice versa.
---

## [Editor Report · Acceptance letter]

31 Aug 2023

PONE-D-22-19835R3 

Changes in anxiety and depression levels and meat intake following recognition of low genetic risk for high body mass index, triglycerides, and lipoproteins: A randomized controlled trial 

Dear Dr. Han:

I'm pleased to inform you that your manuscript has been deemed suitable for publication in PLOS ONE. Congratulations! Your manuscript is now with our production department. 

Kind regards, 

on behalf of

Dr Reindolf Anokye 

Academic Editor

PLOS ONE